# Learning large DAGs is harder than you Think: Many Losses are Minimal for the wrong DAG

**Jonas Seng**
Technical University of Darmstadt
`jonas.seng@tu-darmstadt.de`

**Matej Zečević**
Technical University of Darmstadt
`matej.zecevic@tu-darmstadt.de`

**Devendra Singh Dhami**
Eindhoven University of Technology
`d.s.dhami@tue.nl`

**Kristian Kersting**
Technical University of Darmstadt
Centre for Cognitive Science of TU Darmstadt
Hessian Center for AI (hessian.AI)
German Center for Artificial Intelligence
`kersting@cs.tu-darmstadt.de`

## Abstract

Structure learning is a crucial task in science, especially in fields such as medicine and biology, where the wrong identification of (in)dependencies among random variables can have significant implications. The primary objective of structure learning is to learn a Directed Acyclic Graph (DAG) that represents the underlying probability distribution of the data. Many prominent DAG learners rely on least square losses or log-likelihood losses for optimization. It is well-known from regression models that least square losses are heavily influenced by the scale of the variables. Recently it has been demonstrated that the scale of data also affects performance of structure learning algorithms, though with a strong focus on linear 2-node systems and simulated data. Moving beyond these results, we provide conditions under which square-based losses are minimal for wrong DAGs in $d$-dimensional cases. Furthermore, we also show that scale can impair performance of structure learners if relations among variables are non-linear for both square based and log-likelihood based losses. We confirm our theoretical findings through extensive experiments on synthetic and real-world data.

## 1 Introduction

Given a finite data sample from an unknown probability distribution, structure learning algorithms aim to recover the graphical structure underlying the data generating process that lead to said unknown probability distribution (for an introduction to probabilistic graphical models see (Koller & Friedman, 2009)). The use of a Directed Acyclic Graph (DAG) as a representation of choice is prevalent in many applications, particularly in Bayesian Networks (BN; see (Pearl & Russell, 2000)).The directedness and acyclicity of DAGs provide significant advantages in proving foundational results, including research in causality, see (Pearl, 2009). In DAGs, a node corresponds to a random variable and each edge marks a direct statistical dependence between two random variables. The absence of an edge encodes (in)direct independencies between random variables. Since DAGs compactly encode statistical (in)dependencies, structure learning algorithms learning DAGs are widely deployed throughout Machine Learning applications. Different approaches have been proposed to solve the non-trivial task of recovering the independence structure from a finite data sample. Some use statistical independence-tests to infer a graph, others use score-functions which are being optimized during learning (Mooij et al., 2020; Peters et al., 2017). In recent years, score based methods became appealing with the development of NOTEARS (NT; (Zheng et al., 2018)) a novel score-based structure learning algorithm with a continuous and differentiable DAG-constraint which avoids the combinatorial nature of constructive constraints. Often the score for these methods is chosen s.t. it maximizes the (log-)likelihood of the data given the model. Under the frequently made Gaussian noise assumption, this reduces to minimizing the mean squared error (MSE). To highlight that the MSE is taken w.r.t. the entire graphical model, we call this score *model mean squared error* (MMSE).

Loh & Bühlmann (2014) showed that least square based losses are unsuited for structure identification of DAGs when the variances of exogenous noise variables remain unknown. Reisach et al. (2021) recently had analyzed specifically the behaviour of NT and similar methods and found evidence that they might exploit these variances during structure learning, thus confirming the results of Loh & Bühlmann (2014) in low-dimensional linear cases. However, both works only considered linear dependencies among the variables in low dimensions and square based losses. Also, real world problems usually involve dozens or more variables and many structure learners optimize log-likelihood or similar losses (e.g. BIC, ELBO). Thus investigating conditions under which optimizing MMSE leads to wrong DAGs in $d$-dimensional cases as well as analyzing whether losses like BIC and ELBO are more appropriate choices for structure learning is of high interest for many practitioners. Consequently our contribution will (1) generalize the above mentioned results to $d$-dimensional and non-linear cases, (2) provide exact conditions under which MMSE fails to identify the correct structure and (3) show that all log-likelihood based losses are susceptible to scaling under appropriate assumptions, both theoretically and empirically.

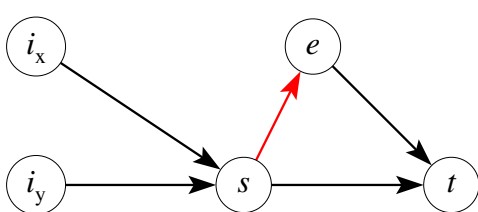

**Medical Example.** To highlight the importance of identifying the correct DAG given data, we provide the following example: Assume a distribution $p_{G_t}$ which is defined as in App. A and factorizes following the Bayesian Network $G_t$ shown in Fig. 1. Further assume we sample data $\mathbf{X}$ from $p_{G_t}$. The variables $i_x, i_y$ denote whether a patient got infected with disease $x$ or $y$ respectively, $s$ denotes if a patient shows symptoms typical for $x$ and $y$, $e$ denotes an exposure to a virus and $t$ denotes if we conduct a treatment. Given two graphs $G_1$ and $G_2$ s.t. $G_1 = G_t$ (i.e., $G_1$ is without red edge in Fig. 1 and $G_2$ with red edge), fitting both

Figure 1: **Network Structure can flip decisions.** Fitting two DAGs $G_1$ (without the red edge) and $G_2$ (with the red edge) on the same data sampled from $p_{G_t}$ changes the probability of assigning a treatment ($t = 1$) given the exact same evidence.

graphs to $\mathbf{X}$ yields different probabilities whether the same patient should receive a treatment: $p_{G_1}(t = 1|i_x = 1, i_y = 0) = 0.49 < 0.51 = p_{G_2}(t = 1|i_x = 1, i_y = 0)$. It is common to choose a threshold value of $0.5$ to decide which action should be taken for variables with a binary domain, thus using $G_2$ instead of $G_1$ would flip the decision from not giving a treatment to giving a treatment based on the *same* evidence. Thus recovering the wrong graph structure can have a severe impact on decision making.

We proceed by stating assumptions made throughout our paper and briefly revisit widely used structure learning algorithms which we use in our experiments. Then, we establish novel theoretical findings on how scale influences MMSE and log-likelihood based scores in structure learning contexts. Finally, we conduct experiments and empirically support our theoretical results before giving a conclusion.

## 2 BACKGROUND AND RELATED WORK

To the best of our knowledge, Loh & Bühlmann (2014) were the first raising the awareness that MSE-like losses are not suitable for the identification of graph structures from data when the variances of exogenous noise variables are unknown. Reisach et al. (2021) recently has shown that structure learning algorithms like NT suffer from performance degradation when data is standardized. Kaiser & Sipos (2021) independently reported similar results. Other important works discussing identifiability and its relation to variance include Peters & Bühlmann (2014); Park (2020); Weichwald et al. (2020).

### 2.1 ASSUMPTIONS

We consider a distribution $p(X_1, \ldots, X_d)$ defined over random variables $X_1, \ldots, X_d$ associated with a DAG $G$ representing the independencies of variables $X_1, \ldots, X_d$ in $p$. The following assumptions are made: **(A1) Immiscible Structures**: $G$ is either single chain, fork or collider with $d$ variables. We also make the common assumption **(A2) Additive Noise Model with Invertable Functions**: Each variable $X_i$ can be written as $X_i = f(\mathbf{PA}_{X_i}^G, \mathcal{N}(0, \sigma))$ where $f$ is a linear or non-linear function that is invertable w.r.t. the noise, $\mathbf{PA}_{X_i}^G$ refers to the parent variables of $X_i$ and $\mathcal{N}(0, \sigma)$ denotes a

zero-mean Gaussian with standard deviation $\sigma$. We assume to obtain $n$ i.i.d. samples from $p$, denoted as $\mathbf{X} \in \mathbb{R}^{n \times d}$.

**Remark 1.** *Note that each DAG is composed of chains, forks and colliders. It is possible that a node $X_i$ of a DAG is e.g. both a fork and a collider. If we do not allow for such structures, each DAG can be decomposed into subgraphs fulfilling **(A1) Immiscible Structures**. This decomposition allows us to reason about far more complex graphs than chains, forks and colliders.*

## 2.2 DETAILS ON DAG LEARNERS INVESTIGATED IN THIS WORK

We briefly present four prominent score based structure learning algorithms which often optimize a square based or log-likelihood based loss. We consider these methods throughout our work as they are widely used and/or constitute the foundation of many more recent algorithms (Lee et al., 2020; Lorch et al., 2021; Wei et al., 2020; Zhang et al., 2022).

**GES (Chickering, 2003).** Greedy Equivalence Search (GES) is a score based structure learner which can identify the correct structure of a Bayesian Network up to Markov equivalence. It consists of two phases: Starting from an empty graph, edges are added to optimize some score/loss $\mathcal{L}$. A second phase aims to remove as many edges as possible until $\mathcal{L}$ cannot be improved anymore as the first phase might add to many edges. Often $\mathcal{L}$ is the Bayesian Information Criterion (BIC). Note that GES does not necessarily learn a DAG as it aims to find Markov Equivalence Classes (MECs). However, having variables on different scales assigns higher scores to certain DAGs in a MEC. We consider the highest scoring DAGs as the solution of GES.

**NT (Zheng et al., 2018).** Given data $\mathbf{X} \in \mathbb{R}^{n \times d}$ from a distribution $p(X_1, \ldots, X_d)$ induced by an unknown DAG $G$, NT aims to recover $G$'s structure by solving a constraint optimization problem where $G$ is represented by a weighted adjacency matrix $G := \mathbf{W} \in \mathbb{R}^{d \times d}$ and the optimization problem is defined as $\arg\min_{\mathbf{W} \in \mathbb{R}^{d \times d}} ||\mathbf{X} - \mathbf{X}\mathbf{W}||_F^2 + \lambda||\mathbf{W}||_1 + \alpha h(\mathbf{W})$. Here, $\mathbf{W}$ is the learnt adjacency representing $G$, $\lambda$ is a regularization parameter to favour sparser graphs, $h$ a continuous and differentiable DAG-constraint and $\alpha$ the Lagrange multiplier. The continuous nature of the optimization problem allows efficient gradient based optimization.

**DAG-GNN (Yu et al., 2019).** DAG-GNN (DG) extends NT to learn structures even if relationships are non-linear as long as they are invertible by stating structure learning as an encoding-decoding problem. The encoder aims to learn the inverse function describing the noise variables $Z = \{Z_1, \ldots, Z_d\}$ as a function of the observed variables $X = \{X_1, \ldots, X_d\}$ while the decoder aims to learn the forward-function describing the relationship between noise $Z$ and variables $X$. Encoder- and decoder-parameters as well as the adjacency $\mathbf{W}$ are learned during ELBO optimization with acyclicity-constraint, resulting in an augmented Lagrangian. For more details refer to App. D.1.

**GraN-DAG (Lachapelle et al., 2020)** Given data $\mathbf{X}$ from random variables $X_1, \ldots, X_d$ GraN-DAG (GND) models relationships among variables via a dedicated neural network (NN) for each $X_j$ given $X_{-j}$ where $X_{-j}$ refers to all variables except $X_j$. By following the so called *neural network paths* through the NNs, the authors derive an adjacency $\mathbf{W}$ based on the parameters of all NNs. Then, inspired by NT, $\mathbf{W}$ – and hence the NN parameters – is constrained s.t. $\mathbf{W}$ is a DAG, i.e. $h(\mathbf{W}) = 0$ which is formulated as a Lagrangian augmented optimization problem, solved via gradient descent.

Note that we only consider score based structure learners in this work since constrained based algorithms are based on independence tests which typically do not depend on variable scales. See App. C.9 for details.

## 3 INFLUENCE OF SCALE ON DAG LEARNERS

In this section we will prove conditions under which structure learners optimizing least square and log-likelihood losses are guaranteed to predict a DAG that differs from the ground truth DAG, sometimes even with different $d$-separation statements. Note that there are exactly three structures that any DAG is composed of: (1) chain structures consist of $d$ single-link structures where one single-link structure is followed by another, i.e. $X_1 \rightarrow X_2 \rightarrow \ldots \rightarrow X_d$, (2) fork structures share the same $d$-separation statements as chains do, however their graph structure is slightly different and is

defined as $X_1 \leftarrow ... \leftarrow X_{i-1} \leftarrow X_i \rightarrow X_{i+1} \rightarrow ... \rightarrow X_d$ and finally (3) colliders are defined as $X_1 \rightarrow X_2 \rightarrow \cdots \rightarrow X_i \leftarrow X_{i+1} \leftarrow \cdots \leftarrow X_d$.

## 3.1 DATA GENERATING PROCESS

For our analysis we make the following assumptions about the data generating process: (1) The data $\mathbf{X}$ contains samples from each variable in a DAG $G$, (2) $\mathbf{X}$ is sampled from a distribution $p_G$ which is Markovian w.r.t. $G$ and (3) each dependence among variables in $G$ can be represented as a function $f : \mathbf{PA}_{X_i} \rightarrow X_i$ that is a linear or non-linear function with additive Gaussian noise where $\mathbf{PA}_{X_i}$ are the parents of a variable $X_i$. Also we assume that the variance of each variable in $\mathbf{X}$ only depends on the unit of measurement. For example, when measuring distances, the scale is determined by whether metres (m) or kilometres (km) are measured (e.g. 1km vs 1000m). As a specific unit (e.g. km) often can be expressed in terms of another unit scaled by a constant factor (e.g. km $= 1000 \cdot$ m), we represent measuring on different scales as a linear scaling operation done on data $\mathbf{X}$: For data samples $\mathbf{X}_i^T$ of variable $X_i$, different measuring units can be expressed as $\mathbf{X}_i^T := c \cdot \mathbf{X}_i^T$. Note that the *structure* of the data generating process is invariant w.r.t. the measurement scale.

## 3.2 EXTENDING VARIANCE SENSITIVITY RESULTS TO $d$ DIMENSIONS

For our theoretical analysis let us define the *Model MSE (MMSE)* of a model $f_{\mathbf{W},\theta}$ parameterized by $\theta$ and respecting (in)dependence structure imposed by adjacency $\mathbf{W}$ representing a DAG as:

**Definition 1.** *Given an adjacency matrix $\mathbf{W} \in \mathbb{R}^{d \times d}$ of a DAG of $d$ nodes, $f_{\mathbf{W},\theta} : \mathbb{R}^d \rightarrow \mathbb{R}^d$ and data $\mathbf{X} \in \mathbb{R}^{n \times d}$ we define the Model MSE (MMSE) as $MMSE(\mathbf{X}, f_{\mathbf{W},\theta}) := \frac{1}{2n}||\mathbf{X} - f_{\mathbf{W},\theta}(\mathbf{X})||_F^2$.*

**Remark 2.** *Note that structure learners use constraints such as that the learned graph is a DAG. Often these constraints are expressed as regularization terms. Throughout this section we assume that all such constraints are fulfilled and only consider the space of valid solutions.*

Note that due to **(A2) Additive Noise Model with Invertable Functions** $f_{\mathbf{W},\theta}$ can be written as a vector of functions $f_{\mathbf{W},\theta} = \langle f_{1,\theta}, \dots, f_{d,\theta} \rangle$ where $\mathbf{X}_j^T = f_{j,\theta}(\mathbf{X}_{P_j})$ holds for index set $P_j = \mathbf{Pa}_{X_j}$ selecting all parents of $X_j$ according to $\mathbf{W}$. We will use this property in our subsequent propositions.

### 3.2.1 SQUARE BASED LOSSES

We start our theoretical analysis by showing that MMSE is scale dependent given **(A2)** holds:

**Proposition 1.** *Given $n$ samples $\mathbf{X} \in \mathbb{R}^{n \times d}$ from a distribution $p(X_1, \dots, X_d)$ over real-valued random variables $X_1, \dots, X_d$, an adjacency $\mathbf{W} \in \mathbb{R}^{d \times d}$ representing a DAG and a function $f_{\mathbf{W},\theta}$ respecting (A2) Additive Noise Model with Invertable Functions, the MMSE is proportional to the sum of MSEs of each node in the graph:*

$$MMSE(\mathbf{X}, \mathbf{W}) = \frac{1}{2n}||\mathbf{X} - f_{\mathbf{W},\theta}(\mathbf{X})||_F^2 \propto \sum_{i \in Z} \text{Var}(\mathbf{X}_i^T) + \sum_{i \in N} MSE(\mathbf{X}_i^T, f_{i,\theta}(\mathbf{X}_{P_i}^T)). \quad (1)$$

In the above proposition, $Z$ denotes the set of node-indices without parents, $N$ denotes the node-indices of nodes with parents and $\text{MSE}(\mathbf{X}_i^T, f_{i,\theta}(\mathbf{X}_{P_i}^T))$ denotes the mean squared error between $\mathbf{X}_i^T$ and $f_{i,\theta}(\mathbf{X}_{P_i}^T)$. A proof can be found in App. B. Since the MMSE contains variance terms, it is susceptible to the scale of variables. Knowing conditions when a minimal MMSE corresponds to a wrong DAG would be crucial to understand the behavior of structure learners. Therefore, let us consider only linear dependencies among variables in $G$, i.e. $f_{\mathbf{W},\theta}(\mathbf{X}) = \mathbf{X}\mathbf{W}$ where $\mathbf{W}$ is a weighted adjacency. Our first key result states that, given a chain graph and data from its associated distribution, the MMSE of a reversed version of said chain is smaller than the ground truth if variables can be sorted by its scale:

**Proposition 2.** *Consider a chain graph $G = X_1 \rightarrow \cdots \rightarrow X_d$ and $G' = X_1 \leftarrow \cdots \leftarrow X_d$, a distribution $p(X_1, \dots, X_d)$ over zero-mean random variables $X_1, \dots, X_d$ induced by $G$ and i.i.d. data $\mathbf{X} \in \mathbb{R}^{n \times d}$ from $p$. If $\text{Var}(\mathbf{X}_1^T) > \cdots > \text{Var}(\mathbf{X}_d^T)$ holds, then $MMSE(\mathbf{X}, G') < MMSE(\mathbf{X}, G)$.*

*Proof (Sketch).* The main observation leading to the proof is that (1) the MMSE of a graph with $d$ variables is the sum of $d$ MSEs and (2) when writing the MSEs in terms of the variances and

covariances of the variables only two things change in the MMSE for a reversed chain: (a) instead of $\text{Var}(\mathbf{X}_1^T)$ only $\text{Var}(\mathbf{X}_d^T)$ is added to the sum of MSEs and (b) the regression coefficients between variables $X_i$ and $X_{i+1}$ are computed as $\frac{\text{Cov}(\mathbf{X}_i^T, \mathbf{X}_{i+1}^T)}{\text{Var}(\mathbf{X}_{i+1}^T)}$ instead of $\frac{\text{Cov}(\mathbf{X}_i^T, \mathbf{X}_{i+1}^T)}{\text{Var}(\mathbf{X}_i^T)}$, rendering them smaller due to the symmetry of covariance. Since the weight's magnitude and the variance of the exogenous variable decide whether the original DAG or the reversed chain receives a smaller MMSE, the reversed graph will receive a lower loss. The full proof is given in App. B. $\qquad\square$

Also, if there is no strict order of the variables in terms of their scale, optimizing MMSE leads to a wrong graph as the following proposition shows:

**Proposition 3.** *Consider a d-dimensional chain graph $G = X_1 \to \cdots \to X_d$, a distribution $p$ factorizing according to $G$ and data $\mathbf{X}$ sampled from $p$ induced by $G$. Then $\text{Var}(\mathbf{X}_1^T) > \text{Var}(\mathbf{X}_d^T) - \sum_{i=1}^{d-1} \frac{\text{Cov}(\mathbf{X}_i^T, \mathbf{X}_{i+1}^T)^2}{\text{Var}(\mathbf{X}_{i+1}^T)} + \frac{\text{Cov}(\mathbf{X}_i^T, \mathbf{X}_{i+1}^T)^2}{\text{Var}(\mathbf{X}_i^T)}$ suffices to make MMSE smaller for the reverse chain.*

*Proof (Sketch).* Due to the symmetry of the covariance and the replacement of $\text{Var}(\mathbf{X}_1^T)$ with $\text{Var}(\mathbf{X}_d^T)$ MMSE will be minimized if the condition $\text{Var}(\mathbf{X}_1^T) > \text{Var}(\mathbf{X}_d^T) - \sum_{i=1}^{d-1} \frac{\text{Cov}(\mathbf{X}_i^T, \mathbf{X}_{i+1}^T)^2}{\text{Var}(\mathbf{X}_{i+1}^T)} + \frac{\text{Cov}(\mathbf{X}_i^T, \mathbf{X}_{i+1}^T)^2}{\text{Var}(\mathbf{X}_i^T)}$ holds. Note that the strong statement of the reverse chain being preferred over *all* other possible DAGs is not made, i.e. only the reverse chain is preferred over the ground truth chain. Still, a flip on e.g. the last edge could yield an even lower MMSE. We give empirical evidence for this conjecture in Sec. 4. A full proof is given in App. B. $\qquad\square$

Similar to how scale dictates that MMSE is minimal for the reversed chain, MMSE gets minimal for a fork structure instead of a chain structure if the conditions derived in the following proposition hold:

**Proposition 4.** *Given a chain graph $G = X_1 \to \cdots \to X_d$ and data $\mathbf{X} \in \mathbb{R}^{n \times d}$ sampled from a distribution $p(X_1, \ldots, X_d)$ induced by $G$, a fork originating at node $X_j$ receives a lower MMSE if the scale of the variables is such that $\text{Var}(\mathbf{X}_i^T) > \text{Var}(\mathbf{X}_{i+1}^T) \quad \forall i \in \{1, \ldots, j-1\}$ and $\text{Var}(\mathbf{X}_i^T) < \text{Var}(\mathbf{X}_{i+1}^T) \quad \forall i \in \{j+1, \ldots, d-1\}$.*

*Proof (Sketch).* The key observation is that one can consider the outgoing paths of $X_j$ as chains. In this case, the right-going path from $X_j$ will receive the same MMSE as in a chain graph if we ignore the rest of the graph. For the left-going path from $X_j$ the same reasoning applies as for reversing a chain-graph then. See App. B for a full proof. $\qquad\square$

Prop. 2-4 can be generalized to a Markov Equivalence Classes (MEC):

**Proposition 5.** *Consider a DAG $G$ with nodes $X_1, \ldots, X_d$, its MEC $\mathcal{M}(G)$ and data $\mathbf{X} \in \mathbb{R}^{n \times d}$ sampled from a distribution $p(X_1, \ldots, X_d)$ induced by $G$. Then the graph $G' \in \mathcal{M}(G)$ will receive a lower MMSE than all other graphs in the same MEC if the edges of $G'$ are oriented s.t. they point from lower-variances variables towards strictly higher-variance variables.*

The proof follows the same reasoning as the more special cases in Prop. 2, 4, see App. B.

**Proposition 6.** *Given a chain $X_1 \to \cdots \to X_d$ and data $\mathbf{X}$ from a distribution $p(X_1, \ldots, X_d)$, a DAG having $X_j$ as a collider and an edge between $X_{j-1}$ and $X_{j+1}$ gets a smaller MMSE than a DAG without additional edge if $\text{Var}(\mathbf{X}_1^T) < \cdots < \text{Var}(\mathbf{X}_j^T)$ and $\text{Var}(\mathbf{X}_{j+1}^T) > \cdots > \text{Var}(\mathbf{X}_d^T)$.*

*Proof (Sketch).* The path $X_1 \to \cdots \to X_{j-1}$ is the same as in the original graph, thus the MSEs are the same. By Prop. 2 a chain $X_d \to \cdots \to X_j$ will receive a lower MSE than vice versa. The additional edge can be explained by the fact that $X_{j-1}$ and $X_{j+1}$ are dependent in $\mathbf{X}$, the collider however would induce an independence between these variables. Thus, to further minimize the MMSE, an edge between $X_{j-1}$ and $X_{j+1}$ is required. See App. B for the full proof. $\qquad\square$

The above proposition shows that DAG learners using optimizing MMSE predict a collider and an additional edge if variables are scaled appropriately, effectively encoding different independencies.

### 3.2.2 LOG-LIKELIHOOD BASED LOSSES

As Prop. 2-6 only apply to the MMSE-loss under the assumption of linear dependencies among variables, we will now generalize our results to non-linear dependencies in two ways: First, we will show that the log-likelihood is as susceptible to scaling as MMSE under Gaussian noise assumption. Second, we generalize this result to log-likelihood *based* losses and show that popular losses such as the Bayesian Information Criterion (BIC) and Evidence Lower Bound (ELBO) are also susceptible to scaling under Gaussian noise assumption:

**Proposition 7.** *Assuming (1) a distribution $p$ over random variables $X_1, \ldots, X_d$ which factorizes according to some DAG $\mathcal{G}$ which is represented by an adjacency $\mathbf{W}$, (2) Gaussian, additive noise $\epsilon_j$ for each $X_j$ in the graph, (3) possibly non-linear transformations $f_j$ s.t. $X_j = f_j(\mathbf{Pa}_{X_j}) + \epsilon_j$, (4) data $\mathbf{X} \in \mathbb{R}^{n \times d}$ sampled from $p$ and (5) a solution space of DAG adjacency matrices $\mathbf{W}$, then optimizing the MMSE is equivalent to optimizing the log-likelihood.*

*Proof (Sketch).* Given data $\mathbf{X}$, the log-likelihood w.r.t. a DAG and parameterized functions $f_{j,\theta}$ for each $X_j$ predicting parameters for a distribution over $X_j$ given $\mathbf{Pa}_{X_j}$ can be recursively written as $\sum_{i=1}^{n} \sum_{j=1}^{d} \log \quad \hat{p}(\mathbf{X}_{i,j}^T | f_{j,\theta}(\mathbf{X}_{P_j}))$ where $P_j = \mathbf{Pa}_{X_j}$ refers to the parents of $X_j$ according to $\mathbf{W}$, $f_{j,\theta}$ refers to a model parameterized by $\theta$ predicting parameters for the distribution $p$ over $X_j$ given its parents. Assuming $\hat{p}$ is Gaussian, the log-likelihood reduces to the MMSE from Definition 1 and reads $\sum_{i=1}^{n} \sum_{j=1}^{d} \left( \mathbf{X}_{i,j}^T - f_{j,\theta}(\mathbf{X}_{P_j}) \right)^2$. Note that $f_{j,\theta}$ can be any function, thus also non-linear. $\square$

We note, however, that in contrast to the linear case, it is not trivial to derive conditions under which the optimum of the log-likelihood render a wrong DAG without additional assumptions about functions $f_{j,\theta}$. We leave the derivation of such conditions for future work.

As the log-likelihood term appears in many scores/losses used for structure learning, we extend Prop. 7 to a more general case. To that end, we define the *family* of log-likelihood based losses as:

**Definition 2.** *The family of log-likelihood losses for a distribution $p$, parameters $\theta$ and a data vector $\mathbf{x} \in \mathbb{R}^n$ is defined as $\mathcal{L}(\mathbf{x}, \theta) = \sum_{i=1}^{n} \log \ p(\mathbf{x}_i | \theta) + h(\cdot)$ where $h(\cdot)$ is some arbitrary function.*

**Proposition 8.** *Under the same assumptions of Prop. 7, the family of log-likelihood losses reduces to square based losses and thus is susceptible to variable scale.*

The proof for Prop. 8 immediately follows from Prop. 7 as it extends the regular log-likelihood by an additive term. We now show that BIC and ELBO as two widely used instantiations of the log-likelihood loss family are susceptible to variable scaling.

**Proposition 9.** *We assume the assumptions of Prop. 7 and further assume an encoder $q(Z|X)$ mapping observed random variables to latents $Z$, decoder $p(X|Z)$ mapping latents $Z$ to observation space $X$ and a prior $p(Z)$. Further, we assume the encoder and decoder to be defined as in Yu et al. (2019). Then, ELBO can be written in terms of the MMSE in Def. 1 and thus is susceptible to scaling.*

*Proof (Sketch).* The ELBO can be written as $-D_{\text{KL}}(q(Z|X)||p(Z)) + \mathbb{E}_{q(Z|X)}\left[ \log(p(X|Z)) \right]$. Here, $Z$ is a multivariate latent and $X$ a multivariate random variable over the input space. Since $\mathbb{E}_{q(Z|X)}\left[ \log(p(X|Z)) \right]$ is an approximation of the log-likelihood of the seen data given the latents and as the log-likelihood w.r.t. a DAG model with adjacency $\mathbf{W}$ can be written as the sum of log-likelihoods of each variable $X_j$ in the DAG, the reconstruction loss of the ELBO reduces to the MMSE due to the Gaussian noise assumption. $\square$

**Proposition 10.** *Under the same assumptions of Prop. 7, BIC can be written in terms of the MMSE in Def. 1 and thus is susceptible to scaling.*

*Proof.* The proof directly follows from Prop. 7 and Def. 2 as BIC w.r.t. an adjacency $\mathbf{W}$ and models $f_{j,\theta}$ can be defined as the sum of the log-likelihood and additive regularization preferring simpler models: $\text{BIC}(\mathbf{X}) = k \ \log(n) - 2\left( \sum_{j=1}^{d} \sum_{i=1}^{n} \log(p(\mathbf{X}_{i,j} | f_{j,\theta}(\mathbf{X}_{P_j}))) \right)$. Here, $k$ refers to the number of model parameters, $n$ the number of samples, the rest as above and $P_j = \mathbf{Pa}_{X_i}$ is an index set selecting the parents of $X_j$. It is easy to see that the BIC is an instantiation of the log-likelihood loss family and hence is susceptible to scaling under Gaussian noise assumption. $\square$

## 3.3 Reducing Variance Sensitivity

As shown above, numerous losses, assuming Gaussian noise, exhibit high sensitivity to the scaling of the measured variables. Free variance terms in losses proportional to MMSE are a major source of misleading information: Nodes lacking parental connections are treated as "not explainable" and their variance is unjustly included in the loss. Hence the graph minimizing free variance terms as far as possible will be rated a good candidate. Also, it is well known that variance influences MSE. However, we argue that a good score should only measure how well *structural dependencies* among variables in a dataset are captured by some DAG and nodes without parents should be treated as unmodelled noise. Consequently, enhancing the resilience of structure learning algorithms can be achieved by: (1) Scaling all variables to have equal variance and (2) excluding free variance terms from the loss, leading to *Scale Robust Loss (SRL)* defined as $\mathcal{L}_s(\mathbf{X}; \mathbf{W}) = \mathcal{L}(\mathbf{X}; \mathbf{W}) - \sum_{i \in Z} \mathrm{Var}(\mathbf{X}_i^T)$. Here, $Z$ is the set of nodes without parents as above and $\mathcal{L}$ a square based or log-likelihood based loss with fixed variance Gaussian noise assumption. Normalization makes MSE independent of scale. By excluding free variances, we ensure that the loss exclusively represents the quality of dependencies encoded by a DAG. Note, however, that SRL can only be used for discrete structure learners as it requires us to know which nodes have no parents. As continuous structure learners operate in a continuous parameter space representing graphs and usually $\mathbf{W}_{ij} \neq 0$ for all $i, j$ in early stages of optimization, each node has at least one parent. The task of devising effective strategies to bolster the resilience of such structure learners against scaling is a matter we defer to future research endeavors.

## 4 Experimental Evaluation

Table 1: **Predictions of structure learners is determined by scale.** We generated 10,000 data points from 10-variable ground truth DAGs (ch=chain, fo=fork or co=collider) with additive Gaussian noise. Variable dependencies were linear (lin) or non-linear (cos=cosine function used). We simulated different measurement scales by multiplying each variable with a different scale, experiments were repeated 30 times with different data from the same distribution to account for stochastic effects. GESR means GES with SRL.

|  |  | Predicted Graph | | |
|---|---|---|---|---|
|  |  | ch | fo | co |
|  | ch | lin (NT) 100% | 100% | 100% |
|  |  | lin (GES) 100% | 100% | 0% |
|  |  | lin (GESR) 50% | 50% | 0% |
|  |  | cos (DG) 100% | 100% | 100% |
|  |  | cos (GND) 100% | 100% | 100% |
|  | fo | lin (NT) 100% | 100% | 100% |
|  |  | lin (GES) 0% | 100% | 100% |
|  |  | lin (GESR) 50% | 0% | 0% |
|  |  | cos (DG) 100% | 100% | 100% |
|  |  | cos (GND) 100% | 100% | 100% |
|  | co | lin (NT) 100% | 100% | 100% |
|  |  | lin (GES) 0% | 0% | 100% |
|  |  | lin (GESR) 0% | 0% | 0% |
|  |  | cos (DG) 100% | 100% | 100% |
|  |  | cos (GND) 100% | 100% | 100% |

(Ground Truth Graph)

After laying the theoretical foundations, we aim to empirically answer the following questions: (**Q1**) Can we empirically confirm our theoretical findings? (**Q2**) How severe are the effects of variable scales if only a subset of variables is measured on different scale? (**Q3**) What happens if assumption (**A1**)[1] **Immiscible Structures** does not hold? (**Q4**) Do scales affect the prediction of structure learners on real world data?

**Confirming theoretical findings (Q1):** To validate our theoretical results, we artificially created data from {3, 10}-node chains, forks and colliders with (non-)linear relations between the variables and additive noise from standard normal distribution (see App. C). We applied scaling operations on each variable in the data to simulate measuring on different scales. Scaling was done to match our assumptions, i.e. the order of variables w.r.t. their variance coincides with a certain DAG we expect the structure learners to predict. For example, in Fig. 2(b)/Fig. 2(c) we sampled data from a chain $X_1 \rightarrow \cdots \rightarrow X_{10}$ and expected a collider/fork at $X_6$ to be predicted due to scaling the variables s.t. the nodes can be sorted by variance according to their topological order with $X_6$ having the largest/smallest variance. Given data from each structure (chain, fork, collider), we tested each possible structure to be predicted by the structure learners due to scaling. For each pair of ground truth, expected prediction and linear/non-linear cases, we used 3 different scales. Each of them was tested 10 times to account for stochastic effects. We

---

[1] We did not relax (**A2**) **Additive Noise Model with Invertable Functions** as it is a common assumption in the literature (Yu et al., 2019).

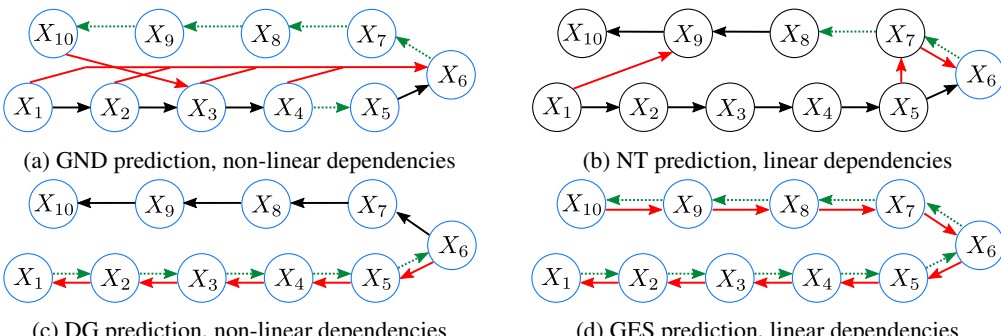

Figure 2: **Predictions of structure learners impaired by measurement scales.** Given a chain as ground truth, NT/GND/DG/GES predict the wrong graph if certain variables are measured on different scales (marked blue). This holds in both, linear (a, c) and non-linear cases (b). The following color coding was used: ⋯⋯▸ means edge in ground truth, → means predicted edge, → means edge appears in ground truth and prediction. (Best viewed in color.)

observed that our expectations were fulfilled in $100\%$ of the cases for NT/DG/GND. Due to its greedy strategy, the prediction of GES did not match our expectation in some cases. However, GES still predicted a graph different to ground truth due to variable scales. SRL improves robustness of GES against scale even further. Thus we empirically confirm our theoretical findings and conclude that scale has severe effects structure learners optimizing a square based loss (see Tab. 1 and App. C).

**Subset of Variables (Q2):** Usually only some variables are measured on different scales. For example, an internet provider could measure the provided download speed in GBit/s and upload speed in MBit/s as often download speed is more important to customers. To investigate how severe the effect of different scales is if just a subset of variables is scaled, we reuse the experimental setup from **Q1**. However, this time scaling was applied to only a subset of variables: Let $\mathbf{X} \in \mathbb{R}^{n \times d}$ denote a dataset with $n$ samples from $d$ variables and $\mathbf{X}(\mathcal{A}) \in \mathbb{R}^{n \times |\mathcal{A}|}$ denote a dataset where a subset $\mathcal{A} \subset \{X_1, \ldots, X_d\}$ of variables was measured on different scale. We conducted experiments with $d \in \{3, 10\}$ for linear and non-linear relationships among the variables. We found that it is enough to measure a single variable on different scale which has at least two neighbor nodes in the ground truth DAG to provoke severe effects. For example, in Fig. 2(b) we expected a collider in $X_6$ to be predicted instead of the ground truth chain only by scaling data from $X_6$ by a factor of 12. We empirically confirmed that all structure learners except GES predict a fork/collider in $100\%$ of the cases if one variable is measured on different scale, for both linear and non-linear dependencies. GES introduces forks/colliders in $\sim 40\%$ of the cases, however often on different nodes.

**Ablation of (A1) Immiscible Structures (Q3):** To see if assumption (**A1**) is strictly required for scale to have severe effects on predictive performance, we generated 20 random DAGs with 10 nodes and artificially generated data as above. None of the random DAGs constituted a single chain, fork or collider, thus violating assumption (A1). We then identified 3-node substructures in each DAG, each making up a chain, fork or collider. We simulated measurements on different scales in these sub-graphs s.t. we expected structure learners to predict a chain/fork/collider-substructure although this substructure is not present in the ground truth. This was done by scaling all variables of the identified substructure (Case 1) or by only scaling one variable (Case 2). Again, each simulation was tested 10 times on data from the same distribution to account for possible stochastic effects. In both cases we found that the identified substructures are severely affected by scale for NT, DG and GND. This holds in $100\%$ of the cases for both linear and non-linear dependencies. For example, in Fig. 3(a) we expected structure learners to predict a collider in $X_1$ instead of the fork structure in the ground truth only because $X_1$ was measured on a scale with factor of 4 larger than the original data. We also discovered that in both cases, many substructures were wrongly recovered without a direct connection to the substructure measured on different scale. We suspect that this is due to the more complex dependence-structure in high dimensional cases. GES was surprisingly robust against scale without SRL, we only encountered changes in the graph's independence statements in $\sim 20\%$ of the cases. The conjecture is that GES benefits from the complex structure and its greedy strategy to find graphs: By greedily searching, graphs being sensitive w.r.t. scale might already be excluded from the candidate set in early stages. We conclude that different scales have severe effects even if (A1) does not hold, however GES seems to be rather robust. For more detailed results refer to App. C.

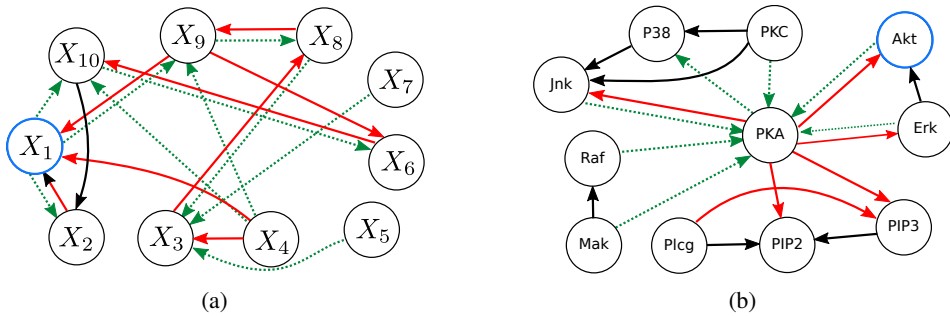

(a)                                           (b)

Figure 3: **Prediction of NT severley impaired by measurement scale complex graphs.** Even in cases with complex dependencies among variables, NT predicts a collider in $X_1$ (Fig. (a)) and *Akt* (Fig. (b)) when data is measured on different scales. (marked blue). Color coding is as in Fig. 2.

**Real World Data (Q4):** Since generated data might not accurately resemble real-world data, we also conduct our experiments with a real world dataset provided in Sachs et al. (2005). As the ground truth graph of this dataset is unknown, the following protocol was used: (1) run a Structure Learning (SL) algorithm on the original data and obtain a graph $G$ predicted by the SL-algorithm, (2) identify a substructure (chain, fork or collider) to alter measurement scale and construct expected graph $G'$, (3) run the same SL-algorithm on the scaled dataset and (4) assess the similarity between the predicted graph $\hat{G}$ after rescaling and $G'$ as well as the similarity between $G$ and the expected graph $G'$. For evaluation: If the substructure we expected to obtain is present in $\hat{G}$, we consider the scale to have severe effect. If this is not the case, the scale is considered non-decisive for the considered substructure. As we know from **Q2** that different measurement scales have unintentional side effects in complex systems, we also measure the similarity $s(G, G')$ and $s(\hat{G}, G')$ where $s$ is the Structural Hamming Distance (SHD). With that we aim to capture the severity of side effects imposed by measurement scale. We again observed that in $100\%$ of our simulations severe effects were present due to scale except for GES, as shown in App. C. Fig. 3(b) illustrates measurement on different scales s.t. *Akt* was turned into a collider. GES, again, is more robust against scale and did predict a chain/fork substructure in $\hat{G}$ if said substructure was a collider in $G$. In Tab. 8, 9 in App. C we provide measures for the severity of side effects using SHD for different structure learners: Graphs predicted on data with different measurement scales have a higher SHD w.r.t. $G'$ although our expectations were fulfilled, i.e. $\hat{G}$ and $G'$ share at least one substructure. Hence other substructures must contribute to the higher SHD, i.e. different measurement scales also lead to worse predictions w.r.t. SHD. To sum up, we have shown that in real world scenarios – where we have no control about assumptions – predictions of structure learners can be severely impaired by measurements on different scales, at the same time leading to a decrease in performance w.r.t. SHD.

## 5   CONCLUSION

We formally proved that measurement scale can decide which DAGs will minimize the MMSE or log-likelihood based losses such as ELBO and BIC. Hence, predictions of structure learning algorithms which make use of these loss functions should be treated with care. We empirically confirmed our theoretical results in extensive experiments using state of the art structure learners and have shown that discrete structure learners are robust against scale when data is normalized and variances are excluded from the loss.

**Limitations and Future Work.** Although our work provides proofs that measurement scale can determine the DAG minimizing MMSE, ELBO and BIC for $d$-dimensional chain-, fork- and collider-structures, a further theoretical analysis of more complex structures with less assumptions – such as in our experiments – would be interesting. Additionally, we did not incorporate other dynamics in the optimization process in our proofs, such as regularization or other hyperparameters controlling the learning algorithm. An interesting next direction can be to check the theoretical properties for other families of losses. Also, deriving scale-independent score functions is of high importance for reliable score based structure learning.

## REPRODUCIBILITY

We acknowledge the significance of reproducibility in scientific research and have taken multiple steps to ensure the strength and replicability of our work.

**Code:** Our implementation is accessible on GitHub at https://github.com/J0nasSeng/FooLS. We have used publicly available software and libraries to guarantee accessibility and have comprehensively described the architecture, software, versions, and hyperparameters in App. C. Our code is deterministic, incorporating seeds for all random number generators to guarantee the replicability of results. We attempted to include most of the code used to create the result tables and figures in this manuscript.

**Datasets:** This study only utilizes publicly available datasets which have been correctly cited. Furthermore, the authors contribute to an open source repository containing all the datasets used in this work, which will be made available upon acceptance.

**Algorithm Details:** We have provided thorough descriptions and formulations of our architecture in the main text, supplemented by additional clarifications, and implementation details in Sec. C, ensuring a clear understanding of our contributions and facilitating reproduction. This documentation is intended to provide researchers with all the necessary information for an accurate replication of our experiments.

## ACKNOWLEDGEMENTS

This work was supported from the National High-Performance Computing project for Computational Engineering Sciences (NHR4CES). Furthermore, it benefited from the Hessian Ministry of Higher Education Research, Science and the Arts (HMWK) via the DEPTH group CAUSE of the Hessian Center for AI (hessian.ai) and the cluster project "The Third Wave of AI". The Eindhoven University of Technology authors received support from their Department of Mathematics and Computer Science and the Eindhoven Artificial Intelligence Systems Institute.

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
