SUPPLEMENTARY MATERIAL TO "LEARNING LARGE DAGS IS HARDER THAN YOU THINK: MANY LOSSES ARE MINIMAL FOR THE WRONG DAG"

## A   RELEVANCE

To see the possible damages induced using a wrong DAG consider the following example. Assume a distribution $p_{G_t}$ factorized as in the Bayesian Network $G_t$ shown in Fig. 4a and assume we sample data $\mathbf{X}$ from the distribution $p_{G_t}$ defined below. The variable $i_x$ denotes if a patient got infected with disease $x$, $i_y$ if the patient got infected with $y$, $s$ denotes if a patient shows symptoms that a typical for $x$ and $y$, $e$ denotes an exposure to a virus and $t$ denotes if we conduct a treatment.

Assume we have given two candidate graphs $G_1$ and $G_2$ s.t. $G_1 = G_t$ and $G_2$ as shown in Fig. 4b. If we fit both graphs to $\mathbf{X}$, we obtain different probabilities for the query $p_{G_1}(t = 1 | i_x = 1, i_y = 0)$: $p_{G_1}(t = 1 | i_x = 1, i_y = 0) = 0.49 < 0.51 = p_{G_2}(t = 1 | i_x = 1, i_y = 0)$. It is common to choose a threshold value of $0.5$ to decide which action should be taken for variables with a binary domain, thus if a structure learning yields $G_2$ instead of $G_1$, this would flip the decision from not giving a treatment to giving a treatment based on the same evidence. Thus our manipulations can have a direct influence on eventually critical decision making processes.

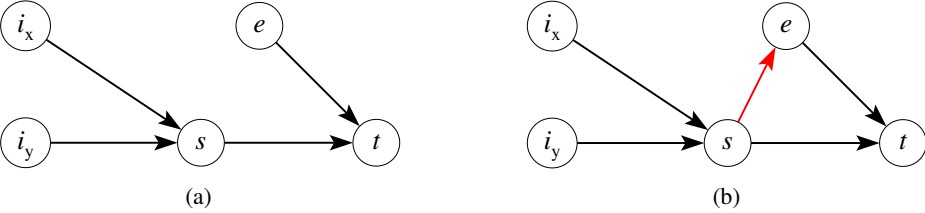

(a)                                                           (b)

Figure 4: **Network Structure can flip decisions.** Figure (a) shows the true independence structure of $p_{G_t}$ in the medical example while (b) shows a wrong independence structure. The probability of assigning a treatment ($t = 1$) is different in both graphs given the exact same evidence.

| $i_x$ | $p(i_x)$ |
|---|---|
| 0 | 0.9 |
| 1 | 0.1 |

| $i_y$ | $p(i_y)$ |
|---|---|
| 0 | 0.95 |
| 1 | 0.05 |

| $e$ | $p(e)$ |
|---|---|
| 0 | 0.95 |
| 1 | 0.05 |

Table 2: Probability distributions of $i_x$, $i_y$ and $e$.

| | | $p(s = 0 | i_x, i_y)$ | $p(s = 1 | i_x, i_y)$ |
|---|---|---|---|
| $i_x = 0$ | $i_y = 0$ | 0.4 | 0.6 |
| | $i_y = 1$ | 0.3 | 0.7 |
| $i_x = 1$ | $i_y = 0$ | 0.2 | 0.8 |
| | $i_y = 1$ | 0.05 | 0.95 |

Table 3: Data-generating probability distribution conditioned on possible realizations of $s$.

| | | $p(t = 0 | s, e)$ | $p(t = 1 | s, e)$ |
|---|---|---|---|
| $s = 0$ | $e = 0$ | 0.95 | 0.05 |
| | $e = 1$ | 0.2 | 0.8 |
| $s = 1$ | $e = 0$ | 0.4 | 0.6 |
| | $e = 1$ | 0. | 1. |

Table 4: Data-generating probability distribution conditioned on possible realizations of $t$.

## B PROOFS

This section provides the actual proofs as detailed versions of the proof sketches for the theoretical results from the main paper.

**Proposition 1**    In the following we will show that the Model-MSE (MMSE) is equal to the sum of the MSE-terms over all variables in a DAG up to rescaling.

Given a distribution $p(X_1, \ldots, X_d)$ over $\mathbb{R}^d$ with zero mean, unit variance and $n$ i.i.d. samples from $p$, denoted as $\mathbf{X} \in \mathbb{R}^{n \times d}$ where $\mathbf{X}_{ij}$ denotes the $i$-th realization of random variable $X_j$ in $\mathbf{X}$, MMSE is defined as:

$$\text{MMSE}(\mathbf{X}, f_{\mathbf{W}, \theta}) := \frac{1}{2n} ||\mathbf{X} - f_{\mathbf{W}, \theta}(\mathbf{X})||_F^2$$

Here, $\mathbf{W}$ is a $d \times d$-dimensional adjacency representing the (in)dependency-structure of $p$ as a DAG. Further, $f_{\mathbf{W}, \theta}(\mathbf{X})$ is a function representing all functional dependencies among variables $X_i$ in the DAG. We assume that $f_{\mathbf{W}, \theta}(\mathbf{X})$ can be represented as a $d$-dimensional vector of functions where each function computes the value of a variable $X_i$, i.e. $\mathbf{X}_i^T = f_{\theta_i}(\mathbf{X}_{P_i}^T)$ for each $i$ where $P_i = \mathbf{Pa}_{X_i}$ is an index set selecting parents of $X_i$ according to $\mathbf{W}$. Writing out the squared Frobenius-norm, which is defined as $||A||_F^2 := \sum_{i=1}^m \sum_{j=1}^n |a_{ij}|^2$ for a $m \times n$ matrix $A$, yields:

$$\text{MMSE}(\mathbf{X}, f_{\mathbf{W}, \theta}) = \frac{1}{2n} \sum_{i=1}^n \sum_{j=1}^d \left| \left( \mathbf{X} - f_{\mathbf{W}, \theta}(\mathbf{X}) \right)_{ij} \right|^2$$

The Frobenius norm thus is just the summation of the squared absolute values of all elements in the matrix. In this case we consider the residual-matrix $\mathbf{X} - f_{\mathbf{W}, \theta}(\mathbf{X})$. Due to the assumption that $f_{\mathbf{W}, \theta}$ can be represented as a vector of functions, sample $i$ of variable $X_j$ is estimated by $f_{\theta_j}(\mathbf{X}_{P_j, i}^T)$ which allows us to rewrite the MMSE:

$$\begin{aligned}
\text{MMSE}(\mathbf{X}, f_{\mathbf{W}, \theta}) &= \frac{1}{2n} \sum_{i=1}^n \sum_{j=1}^d \left| \left( \mathbf{X}_{ij} - f_{\theta_j}(\mathbf{X}_{P_j, i}^T) \right) \right|^2 \\
&= \frac{1}{2n} \sum_{i=1}^n \sum_{j=1}^d \left( \mathbf{X}_{ij} - f_{\theta_j}(\mathbf{X}_{P_j, i}^T) \right)^2 \\
&= \frac{1}{2n} \sum_{j=1}^d ||\mathbf{X}_j^T - f_{\theta_j}(\mathbf{X}_{P_j}^T)||_2^2
\end{aligned}$$

In the last step we used that the MSE of a variable is the same as the squared euclidean distance between the estimated values and the true values. Thus we can express the MMSE as the sum of $d$ independent MSE-terms. The MSE of a fixed variable with $n$ samples $\mathbf{X}_j^T$ from a fixed variable $X_j$ can be written as:

$$\begin{aligned}
\text{MSE}(\mathbf{X}_j^T, f_{\theta_j}(\mathbf{X}_{P_j}^T)) &= \frac{1}{2n} ||\mathbf{X}_j^T - f_{\theta_j}(\mathbf{X}_{P_j}^T)||_2^2 \\
&\propto \sum_{i=1}^n (\mathbf{X}_{ij} - f_{\theta_j}(\mathbf{X}_{P_j, i}^T))^2
\end{aligned}$$

We will omit the term $\frac{1}{2n}$ in the MSE from now on since it has no effect on the solution minimizing the MSE. There are two cases we have to consider: (1) The weight-vector $\mathbf{W}_j^T = \mathbf{0}$, i.e. a node $X_j$ in the graph represented by $\mathbf{W}$ has no parents, and (2) $\mathbf{W}_j^T \neq \mathbf{0}$, that is, a node $X_j$ in has parents in

the graph. In case of (1) we can shorten the MSE to:

$$\text{MSE}(\mathbf{X}_j^T, f_{\theta_j}(\mathbf{X}_{P_j}^T)) \propto \sum_{i=1}^{n} (\mathbf{X}_{ij} - f_{\theta_j}(\mathbf{X}_{P_j,i}^T))^2$$

$$= \sum_{i=1}^{n} \mathbf{X}_{ij}^2$$

$$= \text{Var}(\mathbf{X}_j^T)$$

This is the case as $P_j$ is the empty set, hence $f_{\theta_j}$ does not exist and can be replaced by a constant (0 in our case). In the last step we used that we assume zero mean of all variables. Thus, the MMSE can be expressed as follows:

$$\text{MMSE}(\mathbf{X}, f_{\mathbf{W},\theta}) = \frac{1}{2n} \sum_{j=1}^{d} ||\mathbf{X}_j^T - f_{\theta_j}(\mathbf{X}_{P_j}^T)||_2^2$$

$$\propto \sum_{j=1}^{d} \mathrm{I}_{\mathbf{W}_j^T=0} \text{Var}(\mathbf{X}_j^T)$$

$$+ (1 - \mathrm{I}_{\mathbf{W}_j^T=0}) \text{MSE}(\mathbf{X}_j^T, f_{\theta_j}(\mathbf{X}_{P_j}^T))$$

$$= \sum_{j \in Z} \text{Var}(\mathbf{X}_j^T) + \sum_{j \in N} \text{MSE}(\mathbf{X}_j^T, f_{\theta_j}(\mathbf{X}_{P_j}^T))$$

Here, $\mathrm{I}_{\mathbf{W}_j^T=0}$ is the indicator function which equals 1 iff $\mathbf{W}_j^T = 0$, $Z$ is the set of variable indices for which $\mathbf{W}_j^T = 0$ holds and $N = \{1, \ldots, d\} \setminus Z$. □

**Proposition 2** Consider a chain graph $X_1 \to \cdots \to X_d$, a distribution $p(X_1, \ldots, X_d)$ s.t. all variables have zero mean and that factorizes according to $G$ and $n$ samples from $p$, denoted as $\mathbf{X} \in \mathbb{R}^{n \times d}$. Then measuring variables on different scales $\text{Var}(\mathbf{X}_1^T) > \cdots > \text{Var}(\mathbf{X}_d^T)$ where $\mathbf{X}_j^T$ denotes a vector of $n$ samples from variables $X_j$. We observe that the Model-MSE (MMSE) of a DAG $G$ represented as a weighted adjacency $\mathbf{W}$ is the sum of $d$ MSE-losses where $d$ is the number of variables in $G$. Thus, if $G$ is a chain $X_1 \to \cdots \to X_d$ we can write:

$$\text{MMSE}(\mathbf{W}, \mathbf{X}) \propto \text{Var}(\mathbf{X}_1^T) + \sum_{j=2}^{d} ||\mathbf{X}_j^T - \mathbf{X}\mathbf{W}_j^T||_2^2$$

To avoid clutter in the notation, we will now refer to the learnt weights describing the dependency between a variable $X_j$ and all other variables as $\mathbf{w}_j$. The MMSE then becomes:

$$\text{MMSE}(\mathbf{W}, \mathbf{X}) \propto \text{Var}(\mathbf{X}_1^T) + \sum_{j=2}^{d} \left(\mathbf{X}_j^T - \mathbf{X}\mathbf{w}_j\right)^2$$

Note that we obtain the $\text{Var}(\mathbf{X}_1^T)$ here because $X_1$ has no parents in the chain, i.e. $\mathbf{w}_1 = \mathbf{0}$. This yields $(\mathbf{X}_1^T - \mathbf{0})^2 = \mathbf{X}_1^T\mathbf{X}_1 = \text{Var}(\mathbf{X}_1^T)$ in the MMSE-loss. Since in a chain there will be exactly one entry for each $X_j, j \neq 1$ in $\mathbf{w}_j$ which is not 0. Thus the MMSE reduces to:

$$\text{MMSE}(\mathbf{W}, \mathbf{X}) \propto \text{Var}(\mathbf{X}_1^T) + \sum_{j=2}^{d} \left(\mathbf{X}_j^T - \mathbf{X}_{j-1}^T w_j\right)^2$$

Here $w_j$ denotes the only entry in $\mathbf{w}_j$ which is non-zero. Since it is assumed that all variables have a zero mean, the squared residuals coincide with the variance of residuals:

$$\text{MMSE}(\mathbf{W}, \mathbf{X}) \propto \text{Var}(\mathbf{X}_1^T) + \sum_{j=2}^{d} \text{Var}(\mathbf{X}_j^T - \mathbf{X}_{j-1}^T w_j)$$

Since for two random variables $A$, $B$ and two scalars $a, b$ $\text{Var}(aA - bB) = a^2\text{Var}(A) + b^2\text{Var}(B) - ab\text{Cov}(A, B)$ holds, the MMSE can then be rewritten in terms of the variances and covariances of

the $d$ variables:

$$\text{MMSE}(\mathbf{W}, \mathbf{X}) \propto \text{Var}(\mathbf{X}_1^T) + \sum_{j=2}^{d} w_j^2 \text{Var}(\mathbf{X}_{j-1}^T)$$
$$- 2w_j \text{Cov}(\mathbf{X}_{j-1}^T, \mathbf{X}_j^T) + \text{Var}(\mathbf{X}_j^T)$$

Since we deal with $d$ linear regressions, we can replace the weights by their analytical solution. Between two variables $A$ and $B$ this corresponds to $w = \frac{\text{Cov}(A,B)}{\text{Var}(A)}$. Replacing the weights with their analytical solution yields:

$$\text{Var}(\mathbf{X}_1^T) + \sum_{j=2}^{d} \left( \frac{\text{Cov}(\mathbf{X}_{j-1}^T, \mathbf{X}_j^T)}{\text{Var}(\mathbf{X}_{j-1}^T)} \right)^2 \text{Var}(\mathbf{X}_{j-1}^T)$$
$$- 2\frac{\text{Cov}(\mathbf{X}_{j-1}^T, \mathbf{X}_j^T)^2}{\text{Var}(\mathbf{X}_{j-1}^T)} + \text{Var}(\mathbf{X}_j^T)$$
$$= \text{Var}(\mathbf{X}_1^T) - \sum_{j=2}^{d} \frac{\text{Cov}(\mathbf{X}_{j-1}^T, \mathbf{X}_j^T)^2}{\text{Var}(\mathbf{X}_{j-1}^T)} + \text{Var}(\mathbf{X}_j^T)$$

Assume the variable's variances are sorted s.t. $\text{Var}(\mathbf{X}_j^T) < \text{Var}(\mathbf{X}_{j+1}^T)$ for all $j \in \{1, \dots, d\}$. Further consider the Markov Equivalence Class (MEC) of $G$. As long as the variables are sorted as assumed, any graph in the MEC of $G$ other than $G$ will receive a higher MMSE. Note that in this case the MEC consists of $G$, any graph $X_1 \leftarrow \cdots \leftarrow X_j \rightarrow \cdots \rightarrow X_d$ with arbitrary $X_j$ and the reverse chain $X_1 \leftarrow \cdots \leftarrow X_d$. Due to the sorted variances, any node selected as the only exogenous variable, i.e. the node with no parents, will contribute more to the MMSE since $X_1$ has the lowest variance. Also, note that the covariance is symmetric, i.e. $\text{Cov}(A, B) = \text{Cov}(B, A)$ for two random variables $A$ and $B$. Thus, the only thing that will change in the sum of the MMSE are the denominators. Since $\frac{\text{Cov}(\mathbf{X}_{j-1}^T, \mathbf{X}_j^T)^2}{\text{Var}(\mathbf{X}_{j-1}^T)} > \frac{\text{Cov}(\mathbf{X}_{j-1}^T, \mathbf{X}_j^T)^2}{\text{Var}(\mathbf{X}_j^T)}$ holds for any $j$, for any graph not respecting the order of variances, there will be at least one of these fractions contributing a smaller value than if the fraction respects the order of variances, thus leading to a larger MMSE. To summarize: The MMSE is minimal iff (1) the selected exogenous variable corresponds to the variable with the lowest variance and (2) all edges are oriented s.t. each edge points from a lower-variance variable towards a higher-variance variable. $\square$

**Proposition 3** Consider a chain graph $G = X_1 \rightarrow \cdots \rightarrow X_d$ (represented as adjacency $\mathbf{W}$), a distribution $p(X_1, \dots, X_d)$ s.t. all variables have zero mean and that factorizes according to $G$ and $n$ samples from $p$, denoted as $\mathbf{X} \in \mathbb{R}^{n \times d}$. Then it suffices to change the variance of $\mathbf{X}_1^T$ s.t. $\text{Var}(\mathbf{X}_1^T) > \text{Var}(\mathbf{X}_d^T) - \sum_{i=1}^{d-1} \frac{\text{Cov}(\mathbf{X}_i^T, \mathbf{X}_{i+1}^T)^2}{\text{Var}(\mathbf{X}_{i+1}^T)} + \frac{\text{Cov}(\mathbf{X}_i^T, \mathbf{X}_{i+1}^T)^2}{\text{Var}(\mathbf{X}_i^T)}$, to prefer the reverse chain $G'$ (represented as adjacency $\mathbf{W}'$) in terms of the MMSE. We know that we can represent the MMSE of $\mathbf{W}$ and $\mathbf{W}'$ as follows:

$$\text{MMSE}(\mathbf{W}, \mathbf{X}) \propto \text{Var}(\mathbf{X}_1^T) - \sum_{j=1}^{d-1} \frac{\text{Cov}(\mathbf{X}_{j+1}^T, \mathbf{X}_j^T)^2}{\text{Var}(\mathbf{X}_j^T)}$$
$$\text{MMSE}(\mathbf{W}', \mathbf{X}) \propto \text{Var}(\mathbf{X}_d^T) - \sum_{j=1}^{d-1} \frac{\text{Cov}(\mathbf{X}_{j+1}^T, \mathbf{X}_j^T)^2}{\text{Var}(\mathbf{X}_{j+1}^T)}$$

In order to make $\text{MMSE}(\mathbf{W}, \mathbf{X}) > \text{MMSE}(\mathbf{W}', \mathbf{X})$ true, we just reorder the terms and obtain:
$$\text{MMSE}(\mathbf{W}, \mathbf{X}) > \text{MMSE}(\mathbf{W}', \mathbf{X})$$
$$\iff \text{Var}(\mathbf{X}_1^T) - \sum_{j=1}^{d-1} \frac{\text{Cov}(\mathbf{X}_{j+1}^T, \mathbf{X}_j^T)^2}{\text{Var}(\mathbf{X}_j^T)} > \text{Var}(\mathbf{X}_d^T) - \sum_{j=1}^{d-1} \frac{\text{Cov}(\mathbf{X}_{j+1}^T, \mathbf{X}_j^T)^2}{\text{Var}(\mathbf{X}_{j+1}^T)}$$
$$\iff \text{Var}(\mathbf{X}_1^T) > \text{Var}(\mathbf{X}_d^T) - \sum_{j=1}^{d-1} \frac{\text{Cov}(\mathbf{X}_{j+1}^T, \mathbf{X}_j^T)^2}{\text{Var}(\mathbf{X}_{j+1}^T)} + \sum_{j=1}^{d-1} \frac{\text{Cov}(\mathbf{X}_{j+1}^T, \mathbf{X}_j^T)^2}{\text{Var}(\mathbf{X}_j^T)}$$

$\square$

**Proposition 4** The proof of Prop. 4 follows a similar reasoning as the last proof of Prop. 2. Let's consider data $\mathbf{X} \in \mathbb{R}^{n \times d}$ that is scaled s.t. $\mathrm{Var}(\mathbf{X}_k^T) < \cdots < \mathrm{Var}(\mathbf{X}_1^T)$ and s.t. $\mathrm{Var}(\mathbf{X}_k^T) < \cdots < \mathrm{Var}(\mathbf{X}_d^T)$ for some $k$, and the fork-graph $G$ $X_1 \leftarrow \cdots \leftarrow X_k \rightarrow \cdots \rightarrow X_d$ as well as the weighted adjacency of $G$, denoted by $\mathbf{W}$. Then the MMSE of $G$ can be written as the sum of the MMSEs of the two sub-graphs $X_1 \leftarrow \cdots \leftarrow X_k$ and $X_k \rightarrow \cdots \rightarrow X_d$:

$$\mathrm{MMSE}(\mathbf{W}, \mathbf{X}) \propto \mathrm{Var}(\mathbf{X}_k^T) - \Big( \sum_{j=1}^{k} \frac{\mathrm{Cov}(\mathbf{X}_{j-1}^T, \mathbf{X}_j^T)^2}{\mathrm{Var}(\mathbf{X}_j^T)}$$
$$+ \sum_{j=k+1}^{d} \frac{\mathrm{Cov}(\mathbf{X}_{j-1}^T, \mathbf{X}_j^T)^2}{\mathrm{Var}(\mathbf{X}_{j-1}^T)} \Big)$$

Now the same reasoning applies as in the proof of Prop. 2: Any other graph in the MEC of $G$ other than $G$ will receive a larger MMSE since either the parent-less node changes, yielding a variance term larger than $\mathrm{Var}(\mathbf{X}_k^T)$, or the sum of covariance terms gets larger due to at least one edge not respecting the variance-oder or both. $\square$

**Proposition 5** Given data from a DAG $G$, represented as a weighted adjacency matrix $\mathbf{W} \in \mathbb{R}^{d \times d}$ and $n$ samples $\mathbf{X} \in \mathbb{R}^{n \times d}$ sampled from a distribution $p(X_1, \ldots, X_d)$ that factorizes according to $G$, we say $G$ respects the order of variances iff $X_i \rightarrow X_j \implies \mathrm{Var}(X_i) < \mathrm{Var}(X_j)$ for all nodes $X_i, X_j$ in $G$. Assuming $G$ respects the order of variances, there is no other graph $G'$ in the MEC of $G$ which has a smaller MMSE. As shown above, the MMSE can be written as:

$$\mathrm{MMSE}(\mathbf{W}, \mathbf{X}) \propto \sum_{j \in Z} \mathrm{Var}(\mathbf{X}_j^T) + \sum_{j \in N} \mathrm{MSE}(\mathbf{X}_j^T, \mathbf{X}, \mathbf{W})$$

Here, $Z$ is the set of nodes in $G$ with no parents (exogenous variables) and $N$ is the set of all nodes in $G$ which have parents (endogenous variables). Let's rewrite the MSE for a given node $X_j$ under the assumption that all variables have mean 0:

$$\mathrm{MSE}(\mathbf{X}_j^T, \mathbf{X}, \mathbf{W}) \propto \sum_{i=1}^{n} \Big( \mathbf{X}_{ij} - \Big( \sum_{k=1}^{d} \mathbf{W}_{kj} \mathbf{X}_{ik} \Big) \Big)^2$$
$$= \mathrm{Var}(\mathbf{X}_j^T - \mathbf{W}_j^T \mathbf{X}^T)$$

Since $\mathrm{Var}(A - B) = \mathrm{Var}(A) + \mathrm{Var}(B) - 2\mathrm{Cov}(A, B)$ holds for two random variables $A$, $B$ and since $\mathbf{W}_j^T \mathbf{X}^T$ can be written as $\sum_{k=1}^{d} \mathbf{W}_{kj} \mathbf{X}_k^T$, we obtain:

$$= \mathrm{Var}(\mathbf{X}_j^T) + \mathrm{Var}(\mathbf{W}_j^T \mathbf{X}^T) - 2\mathrm{Cov}(\mathbf{X}_j^T, \mathbf{W}_j^T \mathbf{X}^T)$$
$$= \mathrm{Var}(\mathbf{X}_j^T) + \mathrm{Var}\Big( \sum_{k=1}^{d} \mathbf{W}_{kj} \mathbf{X}_k^T \Big)$$
$$- 2\mathrm{Cov}\Big( \mathbf{X}_j^T, \sum_{k=1}^{d} \mathbf{W}_{kj} \mathbf{X}_k^T \Big)$$

In the above equation we have the sample-variance of a linear combination of random variables. The variance of a linear combination of $N$ random variables is given by: $\mathrm{Var}(\sum_{i=1}^{N} a_i X_i) = \sum_{i=1}^{N} a_i^2 \mathrm{Var}(X_i) + 2 \sum_{1 \neq i < j \neq N} a_i a_j \mathrm{Cov}(X_i, X_j)$, thus we obtain:

$$= \sum_{k=1}^{d} \mathbf{W}_{kj}^2 \mathrm{Var}(\mathbf{X}_k^T) + 2 \sum_{k=1}^{d} \sum_{i=1}^{k-1} \mathbf{W}_{kj} \mathbf{W}_{ij} \mathrm{Cov}(\mathbf{X}_k^T, \mathbf{X}_i^T)$$
$$+ \mathrm{Var}(\mathbf{X}_j^T) - 2 \sum_{k=1}^{d} \mathbf{W}_{kj} \mathrm{Cov}(\mathbf{X}_j^T, \mathbf{X}_k^T)$$

Since this is equivalent as solving $d$ indepdent linear regressions, all $\mathbf{W}_{ij}$ can be replaced by their analytical solution minimizing the MSE, which is given by: $\mathbf{W}_{ij} = \frac{\mathrm{Cov}(\mathbf{X}_i^T, \mathbf{X}_j^T)}{\mathrm{Var}(\mathbf{X}_i^T)}$. Hence we obtain:

$$= \mathrm{Var}(\mathbf{X}_j^T) + \sum_{k=1}^{d} \frac{\mathrm{Cov}(\mathbf{X}_j^T, \mathbf{X}_k^T)^2}{\mathrm{Var}(\mathbf{X}_k^T)} - 2\sum_{k=1}^{d} \frac{\mathrm{Cov}(\mathbf{X}_j^T, \mathbf{X}_k^T)^2}{\mathrm{Var}(\mathbf{X}_k^T)}$$

$$+ 2\sum_{k=1}^{d}\sum_{i=1}^{k-1} \mathbf{W}_{kj}\mathbf{W}_{ij}\mathrm{Cov}(\mathbf{X}_k^T, \mathbf{X}_i^T)$$

$$= \mathrm{Var}(\mathbf{X}_j^T) - \sum_{k=1}^{d} \frac{\mathrm{Cov}(\mathbf{X}_j^T, \mathbf{X}_k^T)^2}{\mathrm{Var}(\mathbf{X}_k^T)}$$

$$+ 2\sum_{k=1}^{d}\sum_{i=1}^{k-1} \mathbf{W}_{kj}\mathbf{W}_{ij}\mathrm{Cov}(\mathbf{X}_k^T, \mathbf{X}_i^T)$$

Let's consider the case in which we flip an edge s.t. the resulting graph $G'$ is in the same MEC as $G$, but it does no longer respect the variance-order. We observe that the last term in the above equation only appears if a node $X_j$ in a graph has multiple incoming edges, i.e. if $X_j$ is a collider. Since we assume $G'$ to be in the MEC of $G$, all collider-structures stay the same, hence we can treat this term as a constant from now on and can omit it. Replacing the MSE in the MMSE by the above equation then yields:

$$\mathrm{MMSE}(\mathbf{W}, \mathbf{X}) \propto \sum_{j \in Z} \mathrm{Var}(\mathbf{X}_j^T) + \sum_{j \in N} \left( \mathrm{Var}(\mathbf{X}_j^T) \right.$$

$$\left. - \sum_{k=1}^{d} \frac{\mathrm{Cov}(\mathbf{X}_j^T, \mathbf{X}_k^T)^2}{\mathrm{Var}(\mathbf{X}_k^T)} \right)$$

As for all exogenous variables $X_j$ the parent set is empty, there will only be 0-entries in the $j$-th column of the adjacency. This allows us to write the above as:

$$\sum_{j=1}^{d} \left( \mathrm{Var}(\mathbf{X}_j^T) - \sum_{k=1}^{d} \frac{\mathrm{Cov}(\mathbf{X}_j^T, \mathbf{X}_k^T)^2}{\mathrm{Var}(\mathbf{X}_k^T)} \right)$$

Now there are two possible outcomes of manipulating $G$ to obtain a $G'$ being in the MEC of $G$: (1) The set $Z$ remains the same or (2) the set $Z$ changes. This means, either edge-flips lead to a different set of exogenous variables or they do not. In case (1) the MMSE will increase if $G'$ no longer respects the variance-order. This is because all terms stay the same except for the one computing the error for variables that now have an incoming instead of an outgoing edge: An edge flip from $X_j \to X_{j+1}$ to $X_j \leftarrow X_{j+1}$ amounts to changing the optimal weight between $X_j$ and $X_{j+1}$ from $\frac{\mathrm{Cov}(\mathbf{X}_j^T, \mathbf{X}_{j+1}^T)^2}{\mathrm{Var}(\mathbf{X}_j^T)}$ to $\frac{\mathrm{Cov}(\mathbf{X}_j^T, \mathbf{X}_{j+1}^T)^2}{\mathrm{Var}(\mathbf{X}_{j+1}^T)}$. Since we assume $\mathrm{Var}(\mathbf{X}_j) < \mathrm{Var}(\mathbf{X}_{j+1})$ it follows that $\frac{\mathrm{Cov}(\mathbf{X}_j^T, \mathbf{X}_{j+1}^T)^2}{\mathrm{Var}(\mathbf{X}_j^T)} > \frac{\mathrm{Cov}(\mathbf{X}_j^T, \mathbf{X}_{j+1}^T)^2}{\mathrm{Var}(\mathbf{X}_{j+1}^T)}$, thus flipping an edge leads to an increased MMSE.

In case (2) the same reasoning applies. Additionally, the exogenous variables are not the same in $G'$ and $G$. Since we assume that $G$ respects the variance-order and $G'$ does not, there will be at least one exogenous variable in $G'$ which is not an exogenous variable in $G$, leading to a higher MMSE for $G'$. □

**Proposition 6** Given data $\mathbf{X}$ from a distribution $p(X_1, \ldots, X_d)$ induced by a DAG $G = X_1 \to \cdots \to X_d$ and assuming that all $X_j$ have mean 0, the MMSE of a graph $G'$ containing a collider at $X_i$ and an additional edge $X_{i-1} \to X_{i+1}$ will receive a smaller MMSE than $G$ if the variances are scaled s.t. $\mathrm{Var}(\mathbf{X}_k^T) < \mathrm{Var}(\mathbf{X}_{k+1}^T)$ for all $k < i-1$ and $\mathrm{Var}(\mathbf{X}_k^T) > \mathrm{Var}(\mathbf{X}_{k+1}^T)$ for all $k \geq i$. Let's consider the MMSE of two adjacencies $\mathbf{W}$ and $\mathbf{W}'$ representing a weighted version of $G$ and

$G'$ respectively, then the MMSEs can be computed as:

$$\text{MMSE}(\mathbf{W}, \mathbf{X}) = \text{Var}(\mathbf{X}_1^T) + \sum_{k=2}^{d} \text{MSE}(\mathbf{X}_k^T, \mathbf{X}, \mathbf{W})$$

$$\text{MMSE}(\mathbf{W}', \mathbf{X}) = \text{Var}(\mathbf{X}_1^T) + \sum_{k=2}^{i-1} \text{MSE}(\mathbf{X}_k^T, \mathbf{X}, \mathbf{W}')$$

$$\text{MSE}(\mathbf{X}_i^T, \mathbf{X}, \mathbf{W}') + \text{MSE}(\mathbf{X}_{i+1}^T, \mathbf{X}, \mathbf{W}')$$

$$+ \sum_{k=i+2}^{d-1} \text{MSE}(\mathbf{X}_k^T, \mathbf{X}, \mathbf{W}') + \text{Var}(\mathbf{X}_d^T)$$

The reason why we decomposed the MMSE of $\mathbf{W}'$ into multiple MSE-terms is that we can exclude a large part of both MMSEs from our consideration since they are equal: Note that in the MMSE of $\mathbf{W}'$ the first line corresponds to the error contributed to the MMSE by the chain $X_1 \rightarrow \cdots \rightarrow X_{i-1}$ and the third line corresponds to the error coming from the chain $X_{i+1} \leftarrow \cdots \leftarrow X_d$. Line 2 corresponds to the error coming from the collider structure itself. When comparing the two MMSEs we can ignore the error coming from the $X_1 \rightarrow \cdots \rightarrow X_{i-1}$ since it will be the same for both $\mathbf{W}$ and $\mathbf{W}'$. Additionally, the chain $X_{i+1} \leftarrow \cdots \leftarrow X_d$ can be considered independently in the MMSE. Thus we can apply Prop. 2 and conclude that this part of the MMSE of $\mathbf{W}'$ will be smaller than the corresponding part of the MMSE of $\mathbf{W}$ since $\text{Var}(\mathbf{X}_k^T) > \text{Var}(\mathbf{X}_{k+1}^T)$ for all $k > i$. It remains to show that the following holds:

$$\text{MSE}(\mathbf{X}_i^T, \mathbf{X}, \mathbf{W}') + \text{MSE}(\mathbf{X}_{i+1}^T, \mathbf{X}, \mathbf{W}')$$
$$< \text{MSE}(\mathbf{X}_i^T, \mathbf{X}, \mathbf{W}) + \text{MSE}(\mathbf{X}_{i+1}^T, \mathbf{X}, \mathbf{W})$$

Since we assume that all variables have mean 0, the MSE coincides with the variance of the residuals, allowing us to write:

$$\text{Var}(\mathbf{X}_i^T - (\mathbf{W}'_{i-1,i}\mathbf{X}_{i-1}^T + \mathbf{W}'_{i+1,i}\mathbf{X}_{i+1}^T))$$
$$+ \text{Var}(\mathbf{X}_{i+1}^T - (\mathbf{W}'_{i-1,i+1}\mathbf{X}_{i-1}^T + \mathbf{W}'_{i+2,i+1}\mathbf{X}_{i+2}^T))$$
$$< \text{Var}(\mathbf{X}_i^T - (\mathbf{W}_{i-1,i}\mathbf{X}_{i-1}^T))$$
$$+ \text{Var}(\mathbf{X}_{i+1}^T - (\mathbf{W}_{i,i+1}\mathbf{X}_i^T))$$

Since the variance of a linear combination is given by $\text{Var}(\sum_{i=1}^{N} a_i X_i) = \sum_{i=1}^{N} a_i^2 \text{Var}(X_i) + 2\sum_{1 \neq i < j \neq N} a_i a_j \text{Cov}(X_i, X_j)$, we decompose the variance terms accordingly:

$$\left( \left( \text{Var}(\mathbf{X}_i^T) + \mathbf{W}'^2_{i-1,i}\text{Var}(\mathbf{X}_{i-1}^T) + \mathbf{W}'^2_{i+1,i}\text{Var}(\mathbf{X}_{i+1}^T) - 2\mathbf{W}'_{i-1,i}\text{Cov}(\mathbf{X}_{i-1}^T, \mathbf{X}_i^T) - 2\mathbf{W}'_{i+1,i}\text{Cov}(\mathbf{X}_{i+1}^T, \mathbf{X}_i^T) \right. \right.$$

$$\left. - 2\mathbf{W}'_{i-1,i}\mathbf{W}'_{i+1,i}\text{Cov}(\mathbf{X}_{i-1}^T, \mathbf{X}_{i+1}^T) \right) + \left( \text{Var}(\mathbf{X}_{i+1}^T) + \mathbf{W}'^2_{i-1,i+1}\text{Var}(\mathbf{X}_{i-1}^T) + \mathbf{W}'^2_{i+2,i+1}\text{Var}(\mathbf{X}_{i+2}^T) \right.$$

$$\left. \left. - 2\mathbf{W}'_{i-1,i+1}\text{Cov}(\mathbf{X}_{i-1}^T, \mathbf{X}_{i+1}^T) - 2\mathbf{W}'_{i+2,i+1}\text{Cov}(\mathbf{X}_{i+2}^T, \mathbf{X}_{i+1}^T) - 2\mathbf{W}'_{i-1,i+1}\mathbf{W}'_{i+2,i+1}\text{Cov}(\mathbf{X}_{i-1}^T, \mathbf{X}_{i+2}^T) \right) \right)$$

$$< \left( \left( \text{Var}(\mathbf{X}_i^T) + \mathbf{W}^2_{i-1,i}\text{Var}(\mathbf{X}_{i-1}^T) - 2\mathbf{W}_{i-1,i}\text{Cov}(\mathbf{X}_i^T, \mathbf{X}_{i-1}^T) \right) \right.$$

$$\left. + \left( \text{Var}(\mathbf{X}_{i+1}^T) + \mathbf{W}^2_{i,i+1}\text{Var}(\mathbf{X}_i^T) - 2\mathbf{W}_{i,i+1}\text{Cov}(\mathbf{X}_i^T, \mathbf{X}_{i+1}^T) \right) \right)$$

Since $\mathbf{W}'_{i-1,i} = \mathbf{W}_{i-1,i} = \frac{\mathrm{Cov}(\mathbf{X}_i^T, \mathbf{X}_{i-1}^T)}{\mathrm{Var}(\mathbf{X}_{i-1}^T)}$, we can exclude the error contributed by the connection $X_{i-1} \to X_i$:

$$
\begin{aligned}
&\Bigg( \mathbf{W}'^2_{i+1,i}\mathrm{Var}(\mathbf{X}_{i+1}^T) - 2\mathbf{W}'_{i+1,i}\mathrm{Cov}(\mathbf{X}_{i+1}^T, \mathbf{X}_i^T) - 2\mathbf{W}'_{i-1,i}\mathbf{W}'_{i+1,i}\mathrm{Cov}(\mathbf{X}_{i-1}^T, \mathbf{X}_{i+1}^T) + \mathbf{W}'^2_{i-1,i+1}\mathrm{Var}(\mathbf{X}_{i-1}^T) \\
&\quad + \mathbf{W}'^2_{i+2,i+1}\mathrm{Var}(\mathbf{X}_{i+2}^T) - 2\mathbf{W}'_{i-1,i+1}\mathrm{Cov}(\mathbf{X}_{i-1}^T, \mathbf{X}_{i+1}^T) - 2\mathbf{W}'_{i+2,i+1}\mathrm{Cov}(\mathbf{X}_{i+2}^T, \mathbf{X}_{i+1}^T) \\
&\quad - 2\mathbf{W}'_{i-1,i+1}\mathbf{W}'_{i+2,i+1}\mathrm{Cov}(\mathbf{X}_{i-1}^T, \mathbf{X}_{i+2}^T) \Bigg) \\
&< \Bigg( \mathbf{W}^2_{i,i+1}\mathrm{Var}(\mathbf{X}_i^T) - 2\mathbf{W}_{i,i+1}\mathrm{Cov}(\mathbf{X}_i^T, \mathbf{X}_{i+1}^T) \Bigg)
\end{aligned}
$$

Replacing the weights with their analytical solution yields:

$$
\begin{aligned}
&\Bigg( \frac{\mathrm{Cov}(\mathbf{X}_{i+1}^T, \mathbf{X}_i^T)^2}{\mathrm{Var}(\mathbf{X}_{i+1}^T)} - 2\frac{\mathrm{Cov}(\mathbf{X}_{i+1}^T, \mathbf{X}_i^T)^2}{\mathrm{Var}(\mathbf{X}_{i+1}^T)} - 2\frac{\mathrm{Cov}(\mathbf{X}_{i-1}^T, \mathbf{X}_i^T)}{\mathrm{Var}(\mathbf{X}_{i-1}^T)} \frac{\mathrm{Cov}(\mathbf{X}_{i+1}^T, \mathbf{X}_i^T)}{\mathrm{Var}(\mathbf{X}_{i+1}^T)}\mathrm{Cov}(\mathbf{X}_{i-1}^T, \mathbf{X}_{i+1}^T) \\
&\quad + \frac{\mathrm{Cov}(\mathbf{X}_{i-1}^T, \mathbf{X}_{i+1}^T)^2}{\mathrm{Var}(\mathbf{X}_{i-1}^T)} + \frac{\mathrm{Cov}(\mathbf{X}_{i+1}^T, \mathbf{X}_{i+2}^T)^2}{\mathrm{Var}(\mathbf{X}_{i+2}^T)} - 2\frac{\mathrm{Cov}(\mathbf{X}_{i-1}^T, \mathbf{X}_{i+1}^T)^2}{\mathrm{Var}(\mathbf{X}_{i-1}^T)} - 2\frac{\mathrm{Cov}(\mathbf{X}_{i+1}^T, \mathbf{X}_{i+2}^T)^2}{\mathrm{Var}(\mathbf{X}_{i+2}^T)} \\
&\quad - 2\frac{\mathrm{Cov}(\mathbf{X}_{i-1}^T, \mathbf{X}_{i+1}^T)}{\mathrm{Var}(\mathbf{X}_{i-1}^T)} \frac{\mathrm{Cov}(\mathbf{X}_{i+1}^T, \mathbf{X}_{i+2}^T)^2}{\mathrm{Var}(\mathbf{X}_{i+2}^T)}\mathrm{Cov}(\mathbf{X}_{i-1}^T, \mathbf{X}_{i+2}^T) \Bigg) \\
&< \Bigg( \frac{\mathrm{Cov}(\mathbf{X}_i^T, \mathbf{X}_{i+1}^T)^2}{\mathrm{Var}(\mathbf{X}_i^T)} - 2\frac{\mathrm{Cov}(\mathbf{X}_i^T, \mathbf{X}_{i+1}^T)^2}{\mathrm{Var}(\mathbf{X}_i^T)} \Bigg)
\end{aligned}
$$

This reduces to:

$$
\begin{aligned}
&\Bigg( -\frac{\mathrm{Cov}(\mathbf{X}_{i+1}^T, \mathbf{X}_i^T)^2}{\mathrm{Var}(\mathbf{X}_{i+1}^T)} - 2\frac{\mathrm{Cov}(\mathbf{X}_{i-1}^T, \mathbf{X}_i^T)}{\mathrm{Var}(\mathbf{X}_{i-1}^T)} \frac{\mathrm{Cov}(\mathbf{X}_{i+1}^T, \mathbf{X}_i^T)}{\mathrm{Var}(\mathbf{X}_{i+1}^T)}\mathrm{Cov}(\mathbf{X}_{i-1}^T, \mathbf{X}_{i+1}^T) - \frac{\mathrm{Cov}(\mathbf{X}_{i-1}^T, \mathbf{X}_{i+1}^T)^2}{\mathrm{Var}(\mathbf{X}_{i-1}^T)} \\
&\quad - \frac{\mathrm{Cov}(\mathbf{X}_{i+1}^T, \mathbf{X}_{i+2}^T)^2}{\mathrm{Var}(\mathbf{X}_{i+2}^T)} - 2\frac{\mathrm{Cov}(\mathbf{X}_{i-1}^T, \mathbf{X}_{i+1}^T)}{\mathrm{Var}(\mathbf{X}_{i-1}^T)} \frac{\mathrm{Cov}(\mathbf{X}_{i+1}^T, \mathbf{X}_{i+2}^T)^2}{\mathrm{Var}(\mathbf{X}_{i+2}^T)}\mathrm{Cov}(\mathbf{X}_{i-1}^T, \mathbf{X}_{i+2}^T) \Bigg) \\
&< \Bigg( -\frac{\mathrm{Cov}(\mathbf{X}_i^T, \mathbf{X}_{i+1}^T)^2}{\mathrm{Var}(\mathbf{X}_i^T)} \Bigg)
\end{aligned}
$$

Since $\mathrm{Var}(\mathbf{X}_{i+1}^T) < \mathrm{Var}(\mathbf{X}_i^T)$ by definition, $-\frac{\mathrm{Cov}(\mathbf{X}_{i+1}^T, \mathbf{X}_i^T)^2}{\mathrm{Var}(\mathbf{X}_{i+1}^T)} < -\frac{\mathrm{Cov}(\mathbf{X}_i^T, \mathbf{X}_{i+1}^T)^2}{\mathrm{Var}(\mathbf{X}_i^T)}$ holds. All other terms on the right hand side of the equation are negative, thus the inequation will hold. $\square$

**Proposition 7** Now we show that the log-likelihood w.r.t. a DAG $\mathbf{W}$ and parameterized functions $f_{j,\theta}$ mapping the values of parents of $X_j$, denoted by $\mathbf{X}_{P_j}$, is the same as the MMSE from Def. 1 up to scaling. The log-likelihood can be written as:

$$
\sum_{i=1}^{n}\sum_{j=1}^{d} \log \quad \hat{p}\big(\mathbf{X}_{i,j}^T | f_{j,\theta}(\mathbf{X}_{i,P_j})\big)
$$

Assuming $\mu_i^{(j)} = f_{j,\theta}(\mathbf{X}_{P_j})$ and fixed noise $\sigma^{(j)}$ for each $X_j$, replacing $\hat{p}$ with the definition of the normal distribution yields:

$$\sum_{i=1}^{n}\sum_{j=1}^{d} \log \frac{1}{\sigma\sqrt{2\pi}} \exp\left(-\frac{1}{2}\left(\frac{\mathbf{X}_{ij} - \mu_i^{(j)}}{\sigma^{(j)}}\right)^2\right)$$

$$\propto \sum_{i=1}^{n}\sum_{j=1}^{d}\left(\mathbf{X}_{ij} - \mu_i^{(j)}\right)^2$$

$$= \sum_{i=1}^{n}\sum_{j=1}^{d}\left(\mathbf{X}_{ij} - f_{j,\theta}(\mathbf{X}_{i,P_j})\right)^2$$

$$= \sum_{j=1}^{d}\mathrm{MSE}(\mathbf{X}_j^T, f_{j,\theta}(\mathbf{X}_{P_j}))$$

By Prop. 1 this equals the MMSE. $\qquad\square$

**Proposition 9** In this section we aim to show that ELBO and MMSE are—under standard assumptions—strongly related, thus showing that ELBO is susceptible to scale-changes of variables as well. The ELBO-loss defined in Yu et al. (2019) reads:

$$\mathcal{L}(\mathbf{W}, \theta_e, \theta_d; \mathbf{X}) = -D_{\mathrm{KL}}(q(Z|X)||p(Z)) + \mathbb{E}_{q(Z|X)}\big[\log(p(X|Z))\big] + \alpha h(\mathbf{W}) + \frac{\rho}{2}|h(\mathbf{W})|^2$$

In the ELBO loss, $q(Z|X)$ denotes a distribution over $Z$ given $X$, a multivariate latent random variable, $p(X|Z)$ a distribution over random variables $X$ which should reflect the distribution $\mathbf{X}$ was sampled from and $p(Z)$ a prior over the latent space (in our case a multivariate Gaussian with independent components). Let's assume the following data-generating model: The data is generated according to a distribution $p$ over random variables $X_1, \ldots, X_d$. It holds that the indpendencies among $X_1, \ldots, X_d$ in $p$ can be represented with a DAG which is represented using the adjacency $\mathbf{W}$. Also, each variable $X_j$ is determined by the value of its parents and some additive, independent noise-term $\epsilon_j$ following a Gaussian, i.e. each $X_j$ can be represented by $X_j = f_j(\mathbf{PA}_j) + \epsilon_j$ where $f$ is some possibly non-linear, invertible function. Further assume that we only consider adjacencies representing a DAG in our solution space, i.e. we can omit $h$ in the loss. Assuming Gaussian latent variables, the KL divergence reads:

$$D_{\mathrm{KL}}(q(Z|X)||p(Z)) = \frac{1}{2}\sum_{j=1}^{d}\sigma_{Z_j}^2 + \mu_{Z_j}^2 - 2\log(\sigma_{Z_j}) - 1$$

In the above equation $\mu_{Z_j}$ denotes the $j$-th component of a mean vector of a Gaussian over $Z$, $\sigma_{Z_j}$ denotes the standard deviation of the Gaussian. Assuming $\sigma_{Z_j} = 1$ for all $j$ leaves us with:

$$D_{\mathrm{KL}}(q(Z|X)||p(Z)) \propto \frac{1}{2}\sum_{j=1}^{d}\mu_{Z_j}^2$$

Proceeding with the reconstruction loss, in Yu et al. (2019) a Monte Carlo approach is used to approximate the reconstruction loss for a given sample $\mathbf{X}_i$, which reads

$$\frac{1}{L}\sum_{l=1}^{L}\sum_{j=1}^{d} -\frac{(\mathbf{X}_{ij} - \mu_{X_j}^{(l)})^2}{2(\sigma_{X_j}^{(l)})^2} - \log(\sigma_{X_j}^{(l)}) - c$$

Here, $\sigma_{X_j}^{(l)}$ refers to the standard deviation predicted by the decoder for the $l$-th Monte Carlo sample, the same applies for $\mu_{X_j}^{(l)}$. $c$ is a constant in the above equation and thus can be ignored since it does not affect the solution of the optimization problem. Assuming that $\sigma_{X_j} = 1$ for all $j$, this is proportional to:

$$\frac{1}{L}\sum_{l=1}^{L}\sum_{j=1}^{d} -(\mathbf{X}_{ij} - \mu_{X_j}^{(l)})^2$$

This means the loss we are optimizating for reads

$$\frac{1}{L}\sum_{l=1}^{L}\Big(\sum_{j=1}^{d}-(\mathbf{X}_{ij}-\mu_{X_j}^{(l)})^2-\frac{1}{2}\sum_{j=1}^{d}(\mu_{Z_j}^{(l)})^2\Big)$$

$$-\frac{1}{L}\sum_{l=1}^{L}\Big(\sum_{j=1}^{d}(\mathbf{X}_{ij}-\mu_{X_j}^{(l)})^2+\frac{1}{2}\sum_{j=1}^{d}(\mu_{Z_j}^{(l)})^2\Big)$$

Since maximizing a loss is the same as minimizing the negative version of the same loss, we will consider the following loss from now on:

$$\frac{1}{L}\sum_{l=1}^{L}\Big(\sum_{j=1}^{d}(\mathbf{X}_{ij}-\mu_{X_j}^{(l)})^2+\frac{1}{2}\sum_{j=1}^{d}(\mu_{Z_j}^{(l)})^2\Big)$$

By the central limit theorem we know that for $L\to\infty$ the sample mean $\frac{1}{L}\mu_{X_j}^{(l)}$ coincides with the true mean $\mu_{X_j}$ for any component $j$, the same holds for $\mu_{Z_j}$. Thus, with infinite many samples we obtain:

$$\sum_{j=1}^{d}(\mathbf{X}_{ij}-\mu_{X_j})^2+\frac{1}{2}\sum_{j=1}^{d}(\mu_{Z_j})^2$$

As the true mean $\mu_{X_j}$ is constant, we can treat the second term as a constant term. As we drop this term, we obtain the following loss which is still proportional to the original loss:

$$\sum_{j=1}^{d}(\mathbf{X}_{ij}-\mu_{X_j})^2$$

As we only consider the loss of one sample $i$ so far, let us extend the above to minimizing the loss w.r.t. all samples where $\mu_{X_j}^{(i)}$ refers to the true mean predicted by the decoder for the $i$-th sample (i.e. assuming $L\to\infty$):

$$\sum_{i=1}^{n}\sum_{j=1}^{d}(\mathbf{X}_{ij}-\mu_{X_j}^{(i)})^2$$

This loss is proportional to the MMSE for arbitrary non-linear dependencies among the variables $X_j$ which proves that DG is susceptible to variance manipulation as well.

Before concluding the proof, let us consider a special case of the above where we assume that all relations are linear and are described with by a weighted adjacency $\mathbf{W}$. As in Yu et al. (2019), we replace the means $\mu_{X_j}, \mu_{Z_j}$ predicted by the decoder and encoder respectively and obtain:

$$\sum_{j=1}^{d}\Big(\mathbf{X}_{ij}-\big(\mathbf{X}(\mathbf{I}-\mathbf{W})\mathbf{X}(\mathbf{I}-\mathbf{W})^{-1}\big)_{ij}\Big)^2+\frac{1}{2}\sum_{j=1}^{d}\Big(\big((\mathbf{I}-\mathbf{W})\mathbf{X}\big)_{ij}\Big)^2$$

Since this is a loss for a single sample $\mathbf{X}_i$ but we optimize over all samples, the full loss is given by:

$$\sum_{i=1}^{n}\Big(\sum_{j=1}^{d}\Big(\mathbf{X}_{ij}-\big(\mathbf{X}(\mathbf{I}-\mathbf{W})\mathbf{X}(\mathbf{I}-\mathbf{W})^{-1}\big)_{ij}\Big)^2+\frac{1}{2}\sum_{j=1}^{d}\Big(\big((\mathbf{I}-\mathbf{W})\mathbf{X}\big)_{ij}\Big)^2\Big)$$

$$=\underbrace{\sum_{i=1}^{n}\sum_{j=1}^{d}\Big(\mathbf{X}_{ij}-\big(\mathbf{X}(\mathbf{I}-\mathbf{W})\mathbf{X}(\mathbf{I}-\mathbf{W})^{-1}\big)_{ij}\Big)^2}_{\text{reconstruction loss}}+\underbrace{\frac{1}{2}\sum_{i=1}^{n}\sum_{j=1}^{d}\Big(\big((\mathbf{X}_{ij}-(\mathbf{X}\mathbf{W})_{ij}\big)^2}_{\text{MMSE = KL divergence}}$$

It can be seen that the MMSE-term appears twice in the loss. Assuming perfect reconstruction, the KL-divergence – which is exactly the MMSE in this case – remains. Thus the MMSE is the distance between the marginal likelihood of obtaining the data and the evidence lower bound. □

## C    Experimental Details

### C.1    Perfect Control

In the following we will consider scenarios in which we are able *to control the scale of all variables* in the data $\mathbf{X}$. This amounts to assuming that we can control on which scale all variables are measured. Hence, "manipulating" scale is equivalent to choosing the scale on which a variable is measured. We coin these scenarios as perfect scenarios as we gain full control over variable scales. As mentioned in Section 3, there are three substructures each DAG consists of: Chains, forks and colliders. We will now briefly describe manipulations on each of these structures.

**Chains**    By controlling the variance of variables in $\mathbf{X}$ chains can be reversed and forks as well as colliders can be introduced. E.g. for a graph $X_1 \to X_2 \to X_3$ and data $\mathbf{X}$ for which $\mathrm{Var}(\mathbf{X}_1^T) > \mathrm{Var}(\mathbf{X}_2^T) > \mathrm{Var}(\mathbf{X}_3^T)$ holds, we can force NT and DG to predict $X_1 \leftarrow X_2 \leftarrow X_3$ by scaling data s.t. $\mathrm{Var}(\mathbf{X}_1^T) < \mathrm{Var}(\mathbf{X}_2^T) < \mathrm{Var}(\mathbf{X}_3^T)$ holds. Similarly we can force both algorithms to predict a fork $X_1 \leftarrow X_2 \to X_3$ instead of a chain $X_1 \to X_2 \to X_3$ by scaling data s.t. $\mathrm{Var}(\mathbf{X}_1^T) < \mathrm{Var}(\mathbf{X}_1^T)$ and $\mathrm{Var}(\mathbf{X}_2^T) < \mathrm{Var}(\mathbf{X}_3^T)$.
If we wish NT/DG to predict a collider instead of a chain-structure, the same approach can be used. However, in this case an additional edge will appear in the graph. Let us consider the 3-node example from above again: In the data-generating process each variable is statistically dependent on each other. If a collider is predicted in $X_2$ although data comes from a chain-structure, predicting the collider would ignore the dependence between $X_1$ and $X_3$. Thus an additional edge is introduced to further minimize MMSE/ELBO by NT/DG.

**Forks**    There are two possible ways of manipulating fork-structures: (1) Converting a fork into a chain and (2) converting a fork into a collider. Note that we cannot change the fork's origin node, that is forcing NT/DG to predict e.g. $X_1 \leftarrow X_3 \to X_2$ if data comes from a structure $X_1 \leftarrow X_2 \to X_3$. The reason for this is that we would have to replace the conditional independence $X_3 \perp\!\!\!\perp X_1 | X_2$ by $X_2 \perp\!\!\!\perp X_3 | X_1$ in the data, which is not allowed by the manipulation-definition.
Converting a fork into a chain is rather straightforward and works again by reordering of variances. The same holds for conversion of forks into colliders. Again, converting a graph $X_1 \leftarrow X_2 \to X_3$ to a collider $X_1 \to X_2 \leftarrow X_3$ will result in adding an additional edge between $X_1$ and $X_3$ to respect all dependencies.

**Colliders**    As for forks, there are two possible ways to manipulate collider-structures: (1) Making a collider-structure a chain and (2) making a collider-structure a fork. As for forks, note that we cannot change the sink node of a collider-structure since this would us require to alter the dependence-properties of the data.
We can follow the same reasoning as in the last subsections to control the output of NT/DG in the case of colliders. However, changing a collider-structure to some other will lead to an additional edge being predicted by NT/DG. Consider the collider $X_1 \to X_2 \leftarrow X_3$. By $d$-separation we have a conditional dependence $X_1 \not\!\perp\!\!\!\perp X_3 | X_2$. However, for both, a chain-structure $X_1 \to X_2 \to X_3$ and a fork-structure $X_1 \leftarrow X_2 \to X_3$ the graph implies a conditional *in*dependence $X_1 \perp\!\!\!\perp X_3 | X_2$. Since these independencies do not hold in the data, NT and DG will account for this to minimize MMSE/ELBO and add an additional edge between $X_1$ and $X_3$.

**Manipulation Scale Strategy**    We note that each structure we consider in our theoretical analysis can be viewed as a set of chains from a purely graphical point of view, i.e. ignoring the probabilistic semantics: A fork can be seen as two chains originating in the same node and a collider can be seen as two chains ending in the same node, but starting in different nodes. Therefore we could derive a simple scaling strategy: Given data $\mathbf{X}$ from some distribution $p$ with ground truth graph $G$ and given a target graph $G'$ defined over the same set of variables as $G$, we identify each each in $G'$ and apply the following strategy: If $G'$ is a fork or a chain, we scale the origin node's variance to $1$. Then, for each chain substructure in $G'$, we traverse along the chain's nodes and recursively rescale the variables: For a variable $X_i$ we apply $c \cdot \mathrm{Var}(\mathbf{X}_{i-1}^T)$ where $c \in \{2, 4, 8\}$. The same strategy was applied for NT and DG.

## C.2 IMPERFECT CONTROL

In real world usually only a subset of variables is measured on a "wrong" scale. This amounts to saying we are only allowed to scale a subset of variables freely. We term such scenarios *imperfect control scenarios* as there is no full control over scale anymore. We now investigate if we still can use our theoretical results and methods to control NT's/DG's output, at least s.t. we partially can reach our desired goal.

In the imperfect scenario we consider a dataset $\mathbf{X} \in \mathbb{R}^{n \times d}$ sampled from a distribution $p$ induced by a DAG $G$ where $n$ is the number of instantiations of $d$ random variables $X_i$. We assume to have access to a subset of features, i.e. $\mathcal{A} \subset \{1, \ldots, d\}$. We denote the data accessible and thus manipulate by $\mathbf{X}(\mathcal{A})$. We will now consider the 3-node case in order to show that some manipulations are still possible under these conditions.

Note that it is sufficient to have control over $n - 1$ variables in the data sampled from $p$ in order to perform any arbitrary manipulation successfully, i.e. this would reflect the perfect setting and will not be part of consideration here.

**Chains** We start by considering cases in which data $\mathbf{X}$ comes from a chain structure $X_1 \rightarrow X_2 \rightarrow X_3$ and we only are allowed to manipulate a subset of variables. If we are allowed to manipulate $\mathbf{X}(\{2\})$, it is possible to make NT/DG predicting a collider on $X_2$. The same manipulation can be performed if we have control over $\mathbf{X}(\{3\})$ since we can scale data s.t. $\mathrm{Var}(\mathbf{X}_2^T) > \mathrm{Var}(\mathbf{X}_3^T)$.

If we are able to manipulate $\mathbf{X}(\{1\})$ it is possible to force NT to reverse the chain under some conditions. The success of the manipulation depends on the regularization parameter $\lambda$ chosen for NT as it can be seen in Fig. 5. However, we can be sure that at least the edge between $X_1$ and $X_2$ will be reversed by this manipulation, i.e. we will at least make NT predict a fork. For DG we haven't found a strong connection between the regularization and the success of the manipulation.

**Forks** In case data comes from a fork-structure $X_1 \leftarrow X_2 \rightarrow X_3$, we can perform the same manipulation as above in order to make NT/DG predicting a collider in $X_2$. If we either control $\mathbf{X}(\{1\})$ or $\mathbf{X}(\{3\})$ we can still force NT/DG to predict a chain-structure instead of a fork-structure. There are no other manipulations possible on forks in the restricted 3-node scenario since either we would have control over $n - 1$ nodes leading to a perfect setting or we would have to change the fork's origin node which is not possible as shown in Section 3.

**Colliders** A similar reasoning as for forks applies to colliders. Assuming data from a collider-structure $X_1 \rightarrow X_2 \leftarrow X_3$, we can make NT/DG predicting a fork with origin node $X_2$ if we have control over $\mathbf{X}(\{2\})$. If we have control over $\mathbf{X}(\{1\})$ or $\mathbf{X}(\{3\})$, we can force NT/DG to predict a chain $X_1 \leftarrow X_2 \leftarrow X_3$ with an additional edge $X_1 \leftarrow X_3$ or a chain $X_1 \rightarrow X_2 \rightarrow X_3$ with an additional edge $X_1 \rightarrow X_3$ respectively. Again, no other manipulation is possible since we would either have the perfect setting or we would have to change the collider's sink-node which is impossible as shown in Section 3.

**Scaling Strategy** In the imperfect scenario we apply a different strategy as we are not allowed to scale each node anymore. In our experiments we only scaled one single variable in the imperfect scenario. Therefore we have to distinguish between three cases: The target graph is a chain, fork or collider structure. In case of a chain, we only scaled the variable corresponding to the chain's sink node by applying $(c + v) \cdot \mathrm{Var}(\mathbf{X}_i)$ where $c$ is the maximum scale among all variables except $X_i$ in the data, $X_i$ is the chain's sink node and $v \in \{1, 2, 4\}$.

If the target graph is a fork, we rescale the fork's origin $X_i$ by $\frac{\mathbf{X}_i^T}{(c+v)}$ where $c$ again is the maximum variance of all variables in the data and $v \in \{1, 2, 3\}$. If the target is a collider structure, we apply the same as for the chain case except that the variable being scaled corresponds to the collider node in the target graph.

**Effect of Measuring a Subset of Variables on Different Scales** We provide additional information on the success rate of our manipulations in imperfect scenarios. As described in Sec. 4 **Q2**, we followed the same experimental protocol as for **Q1** except that we only manipulated a single variable. Tab. 7 and 6 provide more insights as well as Fig. 5.

Table 5: **Predictions of structure learners is determined by scale.** We generated 10,000 data points from 10-variable ground truth DAGs (ch=chain, fo=fork or co=collider) with additive Gaussian noise. Variable dependencies were linear (lin) or non-linear (cos=cosine function used). We simulated different measurement scales by multiplying each variable with a different scale, experiments were repeated 30 times with different data from the same distribution to account for stochastic effects. GESR means GES with SRL.

| Ground Truth Graph | | | Predicted Graph | | |
|---|---|---|---|---|---|
| | | | ch | fo | co |
| | ch | lin (NT) | 100% | 100% | 100% |
| | | lin (GES) | 100% | 100% | 0% |
| | | lin (GESR) | 22% | 23% | 0% |
| | | cos (DG) | 100% | 100% | 100% |
| | | cos (GND) | 100% | 100% | 100% |
| | fo | lin (NT) | 100% | 100% | 100% |
| | | lin (GES) | 100% | 0% | 0% |
| | | lin (GESR) | 11% | 0% | 0% |
| | | cos (DG) | 100% | 100% | 100% |
| | | cos (GND) | 100% | 100% | 100% |
| | co | lin (NT) | 100% | 100% | 100% |
| | | lin (GES) | 28% | 38% | 100% |
| | | lin (GESR) | 10% | 10% | 35% |
| | | cos (DG) | 100% | 100% | 100% |
| | | cos (GND) | 100% | 100% | 100% |

Table 6: **Predictions of structure learners suffer in imperfect scenarios.** In $\{3, 10\}$-variable scenarios, our manipulations still consistently force NT/DG to predict the desired DAG with success rate $100\%$ if we aim to produce a fork or collider. Reversing a chain with access to only one variable is sometimes possible, however, the success rate depends on the manipulation scale and hyperparameters of the DAG learner. Also, we saw that in graphs with more variables the success rate decreased. For our experiments, 10,000 data points were generated from each ground truth DAG (chain, fork, or collider) using Gaussian noise. The variable dependencies were either linear or non-linear (cosine function used). We manipulated one variable with three manipulation scales. All manipulations were repeated 10 times with different data from the same distribution to account for stochastic effects. In 3-variable cases fork origins and collider sinks cannot be changed as this requires new dependencies in the data (hence n.a.). If ground truth and prediction are chain structures, the prediction is the reversed chain.

| Ground Truth Graph | | | Predicted Graph | | | | Predicted Graph | | |
|---|---|---|---|---|---|---|---|---|---|
| | | | ch | fo | co | | ch | fo | co |
| | ch | lin (NT) | 100% | 100% | 100% | lin (NT) | 100% | 100% | 100% |
| | | lin (GES) | 100% | 100% | 0% | lin (GES) | 100% | 0% | 0% |
| | | lin (GESR) | 22% | 23% | 0% | lin (GESR) | 0% | 0% | 0% |
| | | cos (DG) | 100% | 100% | 100% | cos (DG) | 100% | 100% | 100% |
| | | cos (GND) | 100% | 100% | 100% | cos (GND) | 100% | 100% | 100% |
| | fo | lin (NT) | 100% | 100% | 100% | lin (NT) | 100% | 100% | 100% |
| | | lin (GES) | 100% | 0% | 0% | lin (GES) | 75% | 70% | 32% |
| | | lin (GESR) | 11% | 0% | 0% | lin (GESR) | 75% | 50% | 0% |
| | | cos (DG) | 100% | 100% | 100% | cos (DG) | 100% | 100% | 100% |
| | | cos (GND) | 100% | 100% | 100% | cos (GND) | 100% | 100% | 100% |
| | co | lin (NT) | 100% | 100% | 100% | lin (NT) | 100% | 100% | 100% |
| | | lin (GES) | 28% | 38% | 100% | lin (GES) | 0% | 0% | 12% |
| | | lin (GESR) | 10% | 10% | 35% | lin (GESR) | 0% | 0% | 12% |
| | | cos (DG) | 100% | 100% | 100% | cos (DG) | 100% | 100% | 100% |
| | | cos (GND) | 100% | 100% | 100% | cos (GND) | 100% | 100% | 100% |
| | | DAGs with 3 variables | | | | DAGs with 10 variables | | | |

Table 7: **Sensitivity towards Scale depends on hyperparameters.** Considering NT, the success ratios of reversing a chain by our manipulations in the imperfect (3 variables) scenario significantly depends on the choice of the scaling factor used in the manipulations and the regularization term $\lambda$ used. With higher manipulation scale and regularization, the success rate decreases. The dependency to the regularization can be explained by the fact that a higher regularization leads to sparser graphs, thus possibly making NT to omit edges which would make the manipulation successful. The dependency between higher manipulation scales and lower success rates may be due to numerical instability during optimization if the variances of variables get too high. For each pair of manipulation scale and regularization we conducted 10.000 experiments, each with a randomly chosen linear function determining $X_2$ and $X_3$ respectively. The ground truth graph was a chain $X_1 \rightarrow X_2 \rightarrow X_3$ and the target graph was $X_1 \leftarrow X_2 \leftarrow X_3$. As noise we used samples from a standard Gaussian.

|  | | Regularization $\lambda$ | | | |
|---|---|---|---|---|---|
|  | | 0 | 0.01 | 0.1 | 1 |
| Manipulation Scale | 2 | 0.35 | 0.32 | 0.23 | 0. |
|  | 4 | 0.30 | 0.28 | 0.15 | 0. |
|  | 8 | 0.23 | 0.21 | 0.10 | 0. |
|  | 10 | 0.19 | 0.19 | 0.10 | 0. |

Table 8: **Different measurement scales can have severe side effects on real world data.** We find that measuring variables on different scales leads to worse predictions ($\hat{G}$) of NT/DG w.r.t. the expected graph $G'$. $M_1$ and $M_2$ refer to the substructures including {Erk, Akt, PKA} and {PKC, P38, Jnk} respectively in which variables were measured on different scales. $M_1$ was evaluated for NT and DG whereas $M_2$ was only evaluated for DG as NT predicted a slightly different DAG on the original data. *cr* means that our expectation was a reversed chain, *if* means that expected a fork being introduced and *ic* is short for introducing collider.

|  | NT Results | | | | DG Results | | | |
|---|---|---|---|---|---|---|---|---|
|  | $M_1$ | | $M_2$ | | $M_1$ | | $M_2$ | |
|  | $s(G, G')$ | $s(\hat{G}, G')$ | $s(G, G')$ | $s(\hat{G}, G')$ | $s(G, G')$ | $s(\hat{G}, G')$ | $s(G, G')$ | $s(\hat{G}, G')$ |
| cr | 9 | 15 | 5 | 9 | 12 | 18 | 20 | 22 |
| if | 0 | 2 | 1 | 3 | 12 | 14 | – | – |
| ic | 12 | 14 | 5 | 7 | 13 | 15 | 19 | 23 |

Table 9: **Different measurement scales can have severe side effects on real world data.** We find that measuring variables on different scales leads to worse predictions ($\hat{G}$) of GND/GES w.r.t. the expected graph $G'$. $M_1$ and $M_2$ refer to the substructures including {Erk, Akt, PKA} and {PKC, P38, Jnk} respectively in which variables were measured on different scales. However, GES seems to be more robust against scale then GND, thus confirming our former findings. *cr* means that our expectation was a reversed chain, *if* means that expected a fork being introduced and *ic* is short for introducing collider.

|  | GND Results | | | | GES Results | | | |
|---|---|---|---|---|---|---|---|---|
|  | $M_1$ | | $M_2$ | | $M_1$ | | $M_2$ | |
|  | $s(G, G')$ | $s(\hat{G}, G')$ | $s(G, G')$ | $s(\hat{G}, G')$ | $s(G, G')$ | $s(\hat{G}, G')$ | $s(G, G')$ | $s(\hat{G}, G')$ |
| cr | 2 | 28 | 3 | 49 | 2 | 6 | 2 | 1 |
| if | 4 | 26 | 1 | 42 | 4 | 0 | 2 | 1 |
| ic | 0 | 35 | 1 | 41 | 2 | 2 | 2 | 2 |

## C.3 THRESHOLDING

NT and DG have a thresholding parameter, denoted as $\tau$ from here on, which controls how high the minimum strength has to be in order to consider a connection found as an edge. Thus $\tau$ acts as another parameter making the graph found by NT sparse, in addition to the $L^1$-regularization. We have run manipulations in the perfect scenario with different values for $\tau$. We chose $\tau \in \{0.001, 0.01, 0.3, 0.5\}$ and obtained the exact same results as described in Section 3 and Section 4 for NT. For DG higher threhsolding values lead to much sparser graphs. For our experiments we decided to use $\tau = 0.1$ for

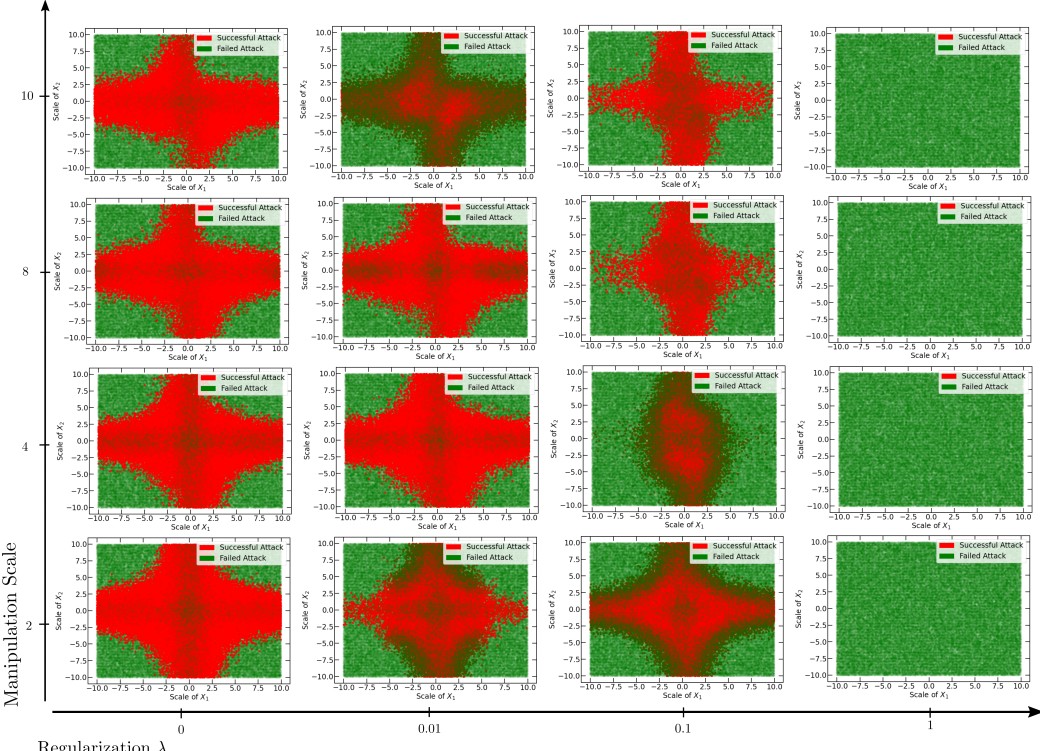

Figure 5: $\lambda$ **and measurement-scale influence success ratio:** In Tab. 7 we provide the overall success rate for each pair of manipulation scale and regulariaztion. Looking at the distribution for which pairs the manipulations are succesful, we find an interesting symmetric pattern which indicates a clear rule underlying the dependency between the regularization/manipulation scale and the success rate.

DG. To conclude, thresholding does not have a significant influence on our results in case of NT, however in case of DG changing $\tau$ might lead to different results.

### C.4 DATA GENERATING PROCESS.

As we require artificial data in some experiments, we describe a general data generating process used: First, define a DAG $G$ and obtain 10.000 samples from a Gaussian distribution with $\mu = 0$ and a standard deviation $\sigma$ for each exogenous variable $X_i$. Each endogenous variable $X_j$ is computed by a function $f$ taking the parents of $X_j$ and adding a Gaussian noise-term $\epsilon_j$, i.e. $X_j = f(\mathbf{PA}_{X_j}) + \epsilon_j$.

### C.5 TECHNICAL DETAILS

Our code is available at https://github.com/J0nasSeng/FooLS. In each manipulation we sampled 10000 samples from a Gaussian distribution for each noise term. Each endogenous node was computed by a linear or non-linear function of its parents and an additive Gaussian noise term. We tested our manipulations using the original NOTEARS implementation (https://github.com/xunzheng/notears), the original DG implementation (https://github.com/fishmoon1234/DAG-GNN), the original GraN-DAG implementation (https://github.com/kurowasan/GraN-DAG) as well as a python implementation of GES (https://github.com/juangamella/ges).
For each experiments in the imperfect manipulation-setting we sampled each noise term only once in order to perform the manipulation on the same data for different manipulation-scales and different values of $\lambda$. The data is available in our repository for reproducibility.
All manipulations were performed on a AMD Ryzen 7 PRO 5850U/Intel Core i7 6700k CPU and 16GB RAM respectively.

Table 10: **Predictions of NT/DG are critically harmed with (A1) Immiscible Structures dropped.** In $\{3, 10\}$-variable scenarios, our manipulations consistently force NT/DG to predict certain sub-structures in the DAG with success rate $100\%$. If ground truth and prediction are chain structures, the prediction is the reversed chain.

| | | | Predicted Graph | | | | | Predicted Graph | | |
|---|---|---|---|---|---|---|---|---|---|---|
| | | | ch | fo | co | | | ch | fo | co |
| | | lin (NT) | 100% | 100% | 100% | | lin (NT) | 100% | 100% | 100% |
| | | lin (GES) | 100% | 100% | 100% | | lin (GES) | 100% | 100% | 100% |
| | ch | lin (GESR) | 0% | 0% | 0% | ch | lin (GESR) | 0% | 0% | 0% |
| | | cos (DG) | 100% | 100% | 100% | | cos (DG) | 100% | 100% | 100% |
| | | cos (GND) | 100% | 100% | 100% | | cos (GND) | 100% | 100% | 100% |
| | | lin (NT) | 100% | 100% | 100% | | lin (NT) | 100% | 100% | 100% |
| | | lin (GES) | 100% | 0% | 100% | | lin (GES) | 100% | 0% | 100% |
| Ground Truth Graph | fo | lin (GESR) | 0% | 0% | 0% | fo | lin (GESR) | 0% | 0% | 0% |
| | | cos (DG) | 100% | 100% | 100% | | cos (DG) | 100% | 100% | 100% |
| | | cos (GND) | 100% | 100% | 100% | | cos (GND) | 100% | 100% | 100% |
| | | lin (NT) | 100% | 100% | 100% | | lin (NT) | 100% | 100% | 100% |
| | | lin (GES) | 100% | 100% | 100% | | lin (GES) | 100% | 100% | 0% |
| | co | lin (GESR) | 0% | 0% | 0% | co | lin (GESR) | 0% | 0% | 0% |
| | | cos (DG) | 100% | 100% | 100% | | cos (DG) | 100% | 100% | 100% |
| | | cos (GND) | 100% | 100% | 100% | | cos (GND) | 100% | 100% | 100% |
| | | All variables scaled | | | | | One variable scaled | | | |

## C.6 Hyperparameters

**NT.** We mostly used standard hyperparameters as in Zheng et al. (2018), i.e. a $L1$-penalty with weight $\lambda = 0.05$, an acyclicity tolerance of $1 \cdot 10^{-8}$ and a threshold of $0.3$ for discretizing the DAG. We allowed a maximum number of $100$ optimization iterations.

**DG.** We used DG mostly with standard hyperparameters as in Yu et al. (2019), i.e. a 2-layer MLP as encoder and decoder. The latent dimension was chosen to have dimension $64$. We optimized using the Adam optimizer with learning rate $0.003$, $\beta = (0.9, 0.999)$, batch size of $100$ and trained for $300$ epochs. The tolerance of the acyclicity constraint was set to $1 \cdot 10^{-8}$.

**GraN-DAG.** We used GraN-DAG with standard hyperparameters except for the pns-threshold which was set to $0.85$ as well as a hard threshold used to filter weak connections from the adjacency (was set to $0.05$ or $0.3$ depending on experiment). The model used was *NonLinGaussANM*.

**GES.** We used standard hyperparameters for GES as well. The only parameter changed was the score for GESR-experiments. Here we used a modified version of BIC in which no free variance terms occur in loss computation.

## C.7 Scale Sensitivity on 10 Nodes with (A1) Immiscible Structures dropped

To examine whether our manipulations still work in $d$-dimensional cases where assumption **(A1) Immiscible Structures** does not hold, we conduct experiments with $d = 10$ nodes and artificially generated data. To do so, we generated 20 random DAGs of which none did constitute a single chain, fork or collider and used this to sample data from it. We used (1) linear functions with random parameters and noise from a standard normal distribution and (2) the cos-function with random parameters and standard Gaussian noise to sample data using the generated DAG. The standard deviation of the noise was set to $1$. After that we applied the manipulation strategy from above with three different scaling factors. We applied NT/DG 10 times for each manipulation scale to account for stochastic effects. We can confirm that our manipulations still work in all cases. However, we obtained that new edges are added or existing edges are removed. The results are shown in Tab. 10.

Table 11: **Measurement scale matters in real world.** Even on the real world dataset where we cannot control/check assumptions, our manipulations can be applied successfully in all cases. We use identifier to refer to certain manipulations as defined in Sec. C.8: The identifiers are of the form (SL M$i${m}). Here, SL is eitehr NT, DG, GraN-DAG or GES and refers to a column. M$i$ refers to a certain substructure and {m} refers to either reversing a chain (rc), introducing a fork (if), introducing a collider (ic) or introducing a chain from another substructure (ich).

|        | NT    | DG    | GraN-DAG | GES   |
|--------|-------|-------|----------|-------|
| M1rc   | 10/10 | 10/10 | 10/10    | 10/10 |
| M1if   | 10/10 | 10/10 | 10/10    | 10/10 |
| M1ic   | 10/10 | 10/10 | 10/10    | 10/10 |
| M2rc   | 10/10 | n.a.  | 10/10    | 10/10 |
| M2if   | 10/10 | n.a.  | 10/10    | 10/10 |
| M2ic   | 10/10 | n.a.  | 10/10    | 10/10 |
| M3ich  | n.a.  | 10/10 | n.a.     | n.a.  |
| M3if   | n.a.  | 10/10 | n.a.     | n.a.  |
| M3ic   | n.a.  | 10/10 | n.a.     | n.a.  |

## C.8 REAL WORLD DATA: SACHS DATASET

We conducted experiments on the dataset of (Sachs et al., 2005) since it is a widely used real-world dataset for evaluation of structure learning algorithms. We ran NT, DG, GraN-DAG and GES on the original and manipulated data to see if the results behave as expected. For NT and GES the following was done: First a pre-processed the dataset was performed by applying a $\log$-transformation on each variable. We applied the variance-manipulations on two sub-structures of the graph to see if our expectations hold. We now briefly describe the manipulations we performed followed by an identifier of the form (SL M$i${m}). Here, SL is eitehr NT or DG, M$i$ refers to a certain substructure and {m} refers to either reversing a chain (rc), introducing a fork (if), introducing a collider (ic) or introducing a chain from another substructure (ich). The identifiers can be used to look up exact results in Tab. 11.

**NT.** In order to reverse the chain Erk $\to$ Akt $\to$ PKA we scaled Erk by 5, Akt by 1.5 and PKA by 0.01 (NT M1cr). To create a fork in Akt we scaled Erk by 5, Akt by 0.5 and PKA by 1.1 (NT M1if). Scaling Erk by 1.5, Akt by 5 and PKA by 0.01 yields a collider in Akt (NT M1ic).
The chain PKC $\to$ P38 $\to$ Jnk was also manipulated: Flipping the chain can be achieved by scaling PKC by 1.1, P38 by 0.3 and Jnk by 0.1 (NT M2rc). Creating a fork in P38 can be achieved by scaling PKC by 3, P38 by 0.3 and Jnk by 1.1 (NT M2if). Scaling PKC by 1.1, P38 by 3 and Jnk by 0.1 yields a collider at P38 (NT M2ic).

**DG.** For DG no pre-processing was performed since it is capable of capturing non-linear dependencies among variables. The same manipulations as for NT were performed. Although DG is susceptible to the manipulations as well, it is harder to find a sufficient scaling factor to flip edges. We suspect that this is due to the non-linearity in the data. Reversing the chain Erk $\to$ Akt $\to$ PKA can be achieved by scaling Erk by 4, Akt by 0.5 and PKA by 0.2 (DG M1rc). Introducing a fork at Akt is achieved by scaling Erk by 4, Akt by 0.5 and PKA by 2 (DG M2if). Creating a collider at Akt can be done by scaling Erk by 0.5, Akt by 2 and PKA by 0.2 (DG M2ic).
The second sub-structure we manipulated in case of DG was Jnk $\leftarrow$ Plcg $\to$ PKA. Converting this fork into a chain was done by scaling Jnk by 2, Plcg by 10 and PKA by 0.01 (DG M3ich). Scaling Jnk by 0.5, Plcg by 10 and PKA by 0.01 yields a collider (DG M3ic). Each manipulation was performed 10 times to account for stochastic effects during optimization. For a summary of our results see Tab. 11.

**GraN-DAG.** The same scaling values of NT were used, see above.

**GES.** As GES is more robust than NT, DG and GraN-DAG, we applied a slightly different strategy: We randomly sampled different scales from a uniform distribution in an interval $[0, 200]$. Each scale-configuration was then tested whether the desired graph was learned. Once configurations fulfilling this criterion were found, we fixed the scales and applied GES on the scaled data 10 times to account for stochastic effects during search.

Table 12: **Constrained-based algorithms are not susceptible to data scale.** We applied the PC algorithm and the Grow Shrink (GS) algorithm on data generated by following the experimental protocol for (Q1). The results clearly show that these two algorithms are not susceptible to data scale.

|  |  | Predicted Graph | | |
|  |  | ch | fo | co |
|---|---|---|---|---|
| Ground Truth Graph | ch | lin (PC) 0% | 0% | 0% |
|  |  | lin (GS) 0% | 0% | 0% |
|  | fo | lin (PC) 0% | 0% | 0% |
|  |  | lin (GS) 0% | 0% | 0% |
|  | co | lin (PC) 0% | 0% | 0% |
|  |  | lin (GS) 0% | 0% | 0% |

### C.9 CONSTRAINED BASED METHODS

Beside score based structure learning, another popular framework for structure learning are constraint based methods which use (conditional) independence tests to identify a graph structure describing the data. As long as the independence test used by a constraint based method is not susceptible to data scale, we do not expect such algorithms to be susceptible neither. This is because independence tests aim to test a null-hypothesis which is either accepted or rejected, depending on whether the data supports the hypothesis or not, i.e. it is not required to compute a score estimating the quality of a graph candidate. We followed the same experimental protocol as in (Q1) and applied the PC algorithm and the Grow Shrink (GS) algorithm on original and scaled data. The results support our conjecture that constraint based methods are not susceptible to data scale, for details refer to App. C.9.

## D DETAILS ON DAG LEARNERS

In this section we provide some details for the DAG learners analyzed in this work.

### D.1 DAG-GNN

DAG-GNN (DG) extends NT to learn structures even if relationships are non-linear as long as they are invertible by stating structure learning as an encoding-decoding problem. The encoder aims to learn the inverse function describing the noise variables $Z = \{Z_1, \ldots, Z_d\}$ as a function of the observed variables $X = \{X_1, \ldots, X_d\}$ while the decoder aims to learn the forward-function describing the relationship between noise $Z$ and variables $X$:

$$
\begin{aligned}
\mu_Z, \log(\sigma_Z) &= (\mathbf{I} - \mathbf{W}^T) f_{\theta_e}(\mathbf{X}) \\
\mu_X, \log(\sigma_X) &= f_{\theta_d}((\mathbf{I} - \mathbf{W}^T)\mathbf{Z})
\end{aligned}
\tag{2}
$$

Here, $\mathbf{I}$ is the identity matrix, $\mathbf{X}$ the data matrix, $\mathbf{W}$ a learnt adjacency matrix, $\mathbf{Z}$ is a sample from $q(Z|X)$, $f_{\theta_e}$ is the encoder network with parameters $\theta_e$, $f_{\theta_d}$ is the decoder network with parameters $\theta_d$, $\mu_Z, \mu_X$ and $\sigma_Z, \sigma_X$ are mean and standard deviation of the learned $q(Z|X)$ and $p(X|Z)$ respectively. Encoder- and decoder-parameters as well as the adjacency $\mathbf{W}$ are learned during ELBO optimization with acyclicity-constraint, resulting in an augmented Lagrangian:

$$
\mathbb{E}_{q(Z|X)}\left[\log(p(X|Z))\right] - D_{\mathrm{KL}}(q(Z|X)||p(Z)) + \alpha h(\mathbf{W}) + \frac{\rho}{2}|h(\mathbf{W})|^2
\tag{3}
$$

In the above equation $D_{\mathrm{KL}}(q(Z|X)||p(Z))$ is the KL divergence between a noise-prior $p(Z)$ and the learnt posterior $q(Z|X)$, $\alpha$ is the Lagrange multiplier and $\rho$ is the penalty parameter.