# OpenReview forum: "Learning Large DAGs is Harder than you Think: Many Losses are Minimal for the Wrong DAG"
_ICLR.cc/2024/Conference — ICLR 2024 poster_

### Official Review · Reviewer_NBUG · 2023-10-23

**Soundness:** 4 excellent
**Presentation:** 3 good
**Contribution:** 2 fair
**Rating:** 6
**Confidence:** 3

**Summary:**

The paper considers many common score/loss functions used in learning DAGs and shows thay can lead to identifying a DAG different than the ground truth. More precisely, the authors show that the (non-)linear score functions are sensitive to the units of measurement used for representing the data, which can at worst lead to the discovered graph encoding an incorrect set of d-separations. This then could lead to bad results in inference. The authors also propose an approach for mitigating the sensitivity to the variance. The theoretical results are complemented by multiple empirical experiments on both synthetic and real-world data to provide evidence of the correctness of the claims.

**Strengths:**

While perhaps not surprising due to structure learning being a notoriously hard problem, the paper presents interesting and to my knowledge novel results on how susceptible many score functions are to scaling and the used units of measurement. One crucial observation the authors make is that poorly chosen units of measurement can lead to the discovered DAG encoding different conditional independencies than which the ground truth DAG encodes. I find this especially important since even finding a DAG from the equivalence class of the ground truth would be sufficient for many problems, but the paper demonstrates that the algorithms may fail to find such a DAG.

The empirical experiments demonstrate well the claims of the paper, i.e., model mean squared error and log-likelihood based losses leading to wrong predictions about the structure of the DAG.

**Weaknesses:**

My main concern is the practicality of the results for the non-linear case; see Questions. I'd be happy to consider increasing my score if the authors offer compelling arguments against my concerns.

Minor + typos:
- Line 5: Add space between "." and "The"
- Table 1 is a bit hard to understand for now. I would add a short explanation in the caption about what the reported probabilities represent (i.e., did the outputted prediction match your expectation).
- Sec. 3.1: The domain of the function $f$ should probably not be the set of parents but instead the power set of the nodes
- End of Sec. 2.2: Missing period after "GraN-DAG (Lachapelle et al., 2020)"
- I'd emphasize that the DAG encodes conditional independencies in Sec. 2, not just independencies

**Questions:**

Could the authors explain how these results relate to the identifiability results of Peters et al. [1, Cor. 31]? They state that the ground truth graph is identifiable assuming that each f is three times differentiable and non-linear, so what is then the motivation of considering MMSE in the non-linear case if it can lead to an incorrect graph as the authors describe?

[1] Jonas Peters, Joris M. Mooij, Dominik Janzing, Bernhard Schölkopf:
Causal discovery with continuous additive noise models. J. Mach. Learn. Res. 15(1): 2009-2053 (2014)

---

> ### Author Response · Authors · 2023-11-13
> **Comment by the Authors**
>
> Thank you for sharing your insightful feedback on our work! We appreciate that you agree with us regarding the importance of our findings. We will of course we will fix all the minor issues you've pointed out, thanks for that! We answer your major concerns below.
>
> > Could the authors explain how these results relate to the identifiability results of Peters et al. [1, Cor. 31]? They state that the ground truth graph is identifiable assuming that each f is three times differentiable and non-linear, so what is then the motivation of considering MMSE in the non-linear case if it can lead to an incorrect graph as the authors describe?
>
> Thank you for sharing this interesting thought! In fact, **we share your concerns** in using log-likelihood based scores (or MMSE-like scores) for structure learning as we explain subsequently:
>
> Indeed, Peters et. al. showed that under the assumption of  three times differentiable, non-linear functions and additive Gaussian noise even the *causal* graph structure is identifiable from a distribution. However, note that the result reported in Peters et. al. are theoretical conditions under which the true DAG does not violate any of the assumptions made (non-linearity in each argument of each function $f_i$ and Gaussian noise) while any other possible DAG would violate these assumptions.
>
> In practice, such a graph is learned by comparing different DAGs and test whether they violate the assumptions or not. Since the number of possible DAGs grows super-exponentially in the number of nodes, one usually only compares DAGs within the same Markov Equivalence Class (MEC) to make the problem feasible, i.e. the set of possible solutions is reduced to graph that share the same independence-structure.
> Identifying the MEC in turn is equivalent to the structure learning tasks our work is considering, i.e. constructing a graph encoding all independencies of the data generating process correctly. However, the methods we considered in our work (except for GES) are supposed to yield one DAG from the correct MEC.
> Identifying the correct MEC or a DAG from a MEC is often done by optimizing log-likelihood (which degenerates to MMSE-like scores under Gaussian noise) due to the following reasons:
> 1. The correct independence structure (i.e. MEC) should be identified if the model found maximizes log-likelihood. This is because wrong independencies would imply a wrong factorization of the corresponding distribution, i.e. yield a worse log-likelihood score.
> 2. Log-likelihood is usually efficient to compute (e.g. additive Gaussian noise case)
>
> Our work shows that under certain conditions log-likelihood is maximized for DAGs encoding wrong independence properties. Hence we share your concerns about using such scores for structure learning tasks.
>
> We hope that we have clarified your concerns and that you can reconsider our rating. Do let us know if you have any further questions or comments.

---

> > ### Comment · Reviewer_NBUG · 2023-11-20
> > **Response to the Authors**
> >
> > I thank the authors for their detailed response to my concerns and improving my understanding of the paper. On the other hand, I agree with the Reviewers kwyp and FNo4 that the presented results are not unexpected, which I consider a minor weakness, although I find their empirical and theoretical verification good.
> >
> > Taking all this into account, I would be inclined to raise my score, but ICLR offers no score option between "marginally above the acceptance threshold" (6) and "accept" (8), so I will have to retain my score. Thus, please read my borderline acceptance more as leaning towards weak acceptance (7).

---

> ### Author Response · Authors · 2023-11-17
> **Looking Forward to Feedback on Our Response**
>
> Dear Reviewer,
>
> We appreciate the time and effort that you have taken to provide us with your insightful review. We would like to ask if the reviewer has any further concerns or is satisfied by our responses to the original review.
>
> We are looking forward to any further discussion with the reviewer and would like to thank the reviewer again for helping make our paper better.
>
> Regards,
>
> The Authors

---

### Official Review · Reviewer_FNo4 · 2023-10-29

**Soundness:** 3 good
**Presentation:** 3 good
**Contribution:** 3 good
**Rating:** 6
**Confidence:** 3

**Summary:**

The paper points out the issue that structural learning approaches that are based on a log likelihood or squared error loss, such as NOTEARS, are typically not scale invariant. This is, these approaches are typically sensitive to the scaling of the data, where rescaling can arbitrarily change the inferred graph structure. The authors provide multiple theoretical statements connecting the variance of the data with the squared loss of a structure learning approach under specific assumptions and evaluate the statements experimentally.

**Strengths:**

- The paper addresses a very important problem and common misconception when log likelihood or squared loss based approaches, like NOTEARS, are used for inferring graph structures.
- Great introduction to the problem.

**Weaknesses:**

- While the implications of the statements are definitely important, they appear rather obvious, since, for instance, it is well known that the L2 loss is closely connected with the variance. In that sense, the issue with the scaling is already discussed in other works. However, that being said, I did not find a paper that particularly focuses on this issue in a self-contained manner.
- The need for A2 remains unclear. In particular, the "invertibility" of the "function" in the related literature typically refers to the invertibility with respect to the noise, not with respect to the input. This is, in a general SEM Y = f(parents, noise), the f needs to be invertible with respect to the noise, but it doesn't matter for the parents. Otherwise, this would be an extremely strong assumption. In the case of additive noise models, this invertibility is always given by definition, since N = Y - f(parents), i.e., it is not a restriction on f. However, I might have also missed something here and maybe the authors can clarify this.
- While you reference great related literature commenting on the same issue, it remains unclear where they lack and where you are filling the gap.
- The proposed approach in Section 3.3 seems rather trivial. Here, I think the main focus of the paper should rely on the theoretical statements.

**Questions:**

The paper points out some very important issues with L2 and log likelihood based approaches. However, many points need more clarification. Some of my questions and remarks, which I hope the authors can address:

- Consider introducing the graph G more formally as a collection of nodes/vertices and edges.
- It would be very insightful to comment more on the assumption A1, since it seems to be very strong. For instance, the formulation of A1 implies that you only consider a chain or a fork or a collider, but none of the combinations. If this is true, this needs a particular remark and justification. Otherwise, it does not seem to be useful for any realistic graph structure beyond these trivial ones.
- In this regard, at the end of paragraph 3, you say that with A1 you consider "all possible substructures", but A1 is clearly defined as "either .." and not a combination.
- Assumption A2 is a bit confusing, since typically, one only requires invertibility with respect to the noise and not the functional relationships (as mentioned in the 'Weakness' section). For instance in DAG-GNN, the "invertibility" only refers to the invertibility of Y = f(X, N) with respect to N. Similar to the previous point,  the need for A2 requires more justification and explanation, since it appears to be a very strong assumption in the current form.
- The unit measurement assumption is a bit unclear. Why is the unit of a node relevant, since, as you also write it, the "rescaling" is part of the functional relationships. In particular, the current paragraph even reads as that these measurements need to be comparable, but looking at the theoretical statements, I don't see why. For instance, one could model "Latency in ms -> Click Rates on a website as discrete number -> Revenue in dollars", where all nodes have a completely different unit.
- While I understand the focus on the MMSE, it should be noted that in an optimization task, some regularization is required. Otherwise, the best solution for ||X - f(X)||_2 is f(X) = X, the identity. Here, a brief remark on this would be insightful to avoid confusion.
- One of my concerns is that the theoretical statements are rather commonly known points. For instance, there is a clear connection between the least squares error and the conditional variance, which is clearly not scale invariant. However, as mentioned before, I still see the value in bringing all these points together in a single paper in the context of structure learning.
- In Section 3.3, the exclusion of "free variance terms" needs more clarification. It currently reads as that one needs to know the graph structure to identify these terms, while the whole task is to infer the graph structure in the first place. I assume there is a misunderstanding where the authors can maybe comment on.
- With regards to the previous point, maybe looking at the conditional variance instead of the marginal variance could help in focusing on the structural connection.

---

> ### Author Response · Authors · 2023-11-13
> **Comment by the Authors (1/2)**
>
> We thank you for your insightful feedback, this really helps us improving our paper. Also we appreciate that you find the presented problem important for structure learning. We answer your concerns below.
>
> >While the implications of the statements are definitely important, they appear rather obvious, since, for instance, it is well known that the L2 loss is closely connected with the variance. In that sense, the issue with the scaling is already discussed in other works. However, that being said, I did not find a paper that particularly focuses on this issue in a self-contained manner.
>
> Thank you for appreciating the importance of our work! We are aware of other works showing or indicating similar problems of e.g. the L2 loss (see [1, 2]). However, these works so far lacked general statements such as our finding that **all** log-likelihood based losses share susceptibility towards scaling in the context of structure learning. Further, we extended existing results to $d$-dimensional cases, hence our results are far more impactful for practical applications. Works like [1, 2] focused on either bi-variate cases, special losses like L2 or both.
>
> >The need for A2 remains unclear. In particular, the "invertibility" of the "function" in the related literature typically refers to the invertibility with respect to the noise, not with respect to the input. This is, in a general SEM Y = f(parents, noise), the f needs to be invertible with respect to the noise, but it doesn't matter for the parents. Otherwise, this would be an extremely strong assumption. In the case of additive noise models, this invertibility is always given by definition, since N = Y - f(parents), i.e., it is not a restriction on f. However, I might have also missed something here and maybe the authors can clarify this.
>
> We thank you for pointing this out! Of course the invertibility only needs to hold w.r.t. the noise. We corrected this in our paper and made Assumption 2 clearer in that regard. However, please note that this does not affect our general theoretical results as A2 is only a requirement for encoder-decoder based structure learners to allow reconstruction.
>
> > While you reference great related literature commenting on the same issue, it remains unclear where they lack and where you are filling the gap.
>
> Let us try to clarify our contributions better: Current works as [1, 2] only discuss similar results for either (1) special loss functions like L2, (2) bi-variate cases, (3) linear dependencies among variables or any combination of these. We fill this gap by presenting clear theoretical results showing that (1) the family of log-likelihood based losses is not suitable for structure learning in (2) $d$-dimensional, (3) non-linear cases. Please note, that we stated our contributions in the Introduction which we now highlighted better in the text:
>
> Consequently our contribution will (1) generalize the above mentioned results to $d$-dimensional and non-linear cases, (2) provide exact conditions under which MMSE fails to identify the correct structure and (3) show that all log-likelihood based losses are susceptible to scaling under appropriate assumptions, both theoretically and empirically.
>
>
> > The proposed approach in Section 3.3 seems rather trivial. Here, I think the main focus of the paper should rely on the theoretical statements.
>
> We agree that the main focus of our paper are the theoretical and empirical results. Section 3.3 is intended as a starting point to tackle the problems of log-likelihood based losses in the context of structure learning and to encourage further research on this. To make this clearer, we have adapted Sec. 3.3 accordingly.
>
> [1] Loh et. al. High-dimensional learning of linear causal networks via inverse covariance estimation. JMLR. 2014.
>
> [2] Reisach et. al. Beware of the simulated dag! varsortability in additive noise models. NeurIPS. 2021.

---

> ### Author Response · Authors · 2023-11-13
> **Comment by the Authors (2/2)**
>
> ## Questions
> > It would be very insightful to comment more on the assumption A1, since it seems to be very strong. For instance, the formulation of A1 implies that you only consider a chain or a fork or a collider, but none of the combinations. If this is true, this needs a particular remark and justification. Otherwise, it does not seem to be useful for any realistic graph structure beyond these trivial ones.
>
> You raise an important point and we agree that an additional remark is beneficial for clarity. In our theoretical analysis we indeed assume chains, forks or colliders. However, note that any DAG is composed of these three structures, hence our results still hold for any arbitrary subgraph where A1 is fulfilled.
> In our point of view, the more important point about A1 is that it implies that each node in a DAG cannot have two roles, e.g. a collider's sink and a fork's origin. Indeed, such cases are not covered by our theoretical findings and are an interesting path for further investigation. We clarified this with an additional remark on A1 in the paper.
>
> > In this regard, at the end of paragraph 3, you say that with A1 you consider "all possible substructures", but A1 is clearly defined as "either .." and not a combination.
>
> What we mean here is that our theoretical results cover all possible substructures a DAG can be composed of except when a node has multiple roles as described above. We clarified this by saying that if A1 holds, each DAG we consider can be decomposed into a set of sub-graphs consisting of chains, forks and colliders.
>
> > The unit measurement assumption is a bit unclear. Why is the unit of a node relevant, since, as you also write it, the "rescaling" is part of the functional relationships. In particular, the current paragraph even reads as that these measurements need to be comparable, but looking at the theoretical statements, I don't see why. For instance, one could model "Latency in ms -> Click Rates on a website as discrete number -> Revenue in dollars", where all nodes have a completely different unit.
>
> In this paragraph we aimed to clarify that the structure of the data generating process is invariant to how we measure variables in terms of scale. For example, if we model the dependence between height and air pressure, the dependence still holds, regardless whether we measure height in meters or kilometres. We included this example in the paper to make our statement clear.
>
> > While I understand the focus on the MMSE, it should be noted that in an optimization task, some regularization is required. Otherwise, the best solution for ||X - f(X)||_2 is f(X) = X, the identity. Here, a brief remark on this would be insightful to avoid confusion.
>
> We fully agree on that and now state explicitly that we only consider adjacency matrices representing a DAG in the beginning of Sec. 3. This is the same as assuming that the optimization procedure is capable of enforcing the constraints represented by regularization terms (e.g. the resulting graph is a DAG).
>
> > In Section 3.3, the exclusion of "free variance terms" needs more clarification. It currently reads as that one needs to know the graph structure to identify these terms, while the whole task is to infer the graph structure in the first place. I assume there is a misunderstanding where the authors can maybe comment on.
>
> Thanks for helping us to clarify this point. Please note that we explicitly state that our proposal of excluding free variance terms from the score only works for discrete optimization algorithms that iterate over candidate graphs (such as GES):
>
> "Consequently, we propose a straightforward protocol for enhancing the resilience of discrete structure learning algorithms, such as the GES algorithm:"
>
> Here, the structure of each candidate graph is known and thus the free variance terms can be excluded from scoring.
>
> > With regards to the previous point, maybe looking at the conditional variance instead of the marginal variance could help in focusing on the structural connection.
>
> Thank you for sharing your thoughts on this! This indeed sounds like a promising idea that is worth being investigated further. We believe that  this might be a similar way as we proposed in Sec. 3 as we exclude free variance terms from the score computation.
>
> We hope that we have clarified your concerns and that you can reconsider our rating. Do let us know if you have any further questions or comments.

---

> ### Author Response · Authors · 2023-11-17
> **Looking Forward to Feedback on Our Response**
>
> Dear Reviewer,
>
> We appreciate the time and effort that you have taken to provide us with your insightful review. We would like to ask if the reviewer has any further concerns or is satisfied by our responses to the original review.
>
> We are looking forward to any further discussion with the reviewer and would like to thank the reviewer again for helping make our paper better.
>
> Regards,
>
> The Authors

---

> > ### Comment · Reviewer_FNo4 · 2023-11-20
> >
> > I want to thank the authors for their thoughtful and detailed response. Most of my concerns were addressed, and I am willing to increase the rating.

---

### Official Review · Reviewer_kwyp · 2023-10-30

**Soundness:** 3 good
**Presentation:** 3 good
**Contribution:** 2 fair
**Rating:** 5
**Confidence:** 3

**Summary:**

In this paper, the authors provide conditions under which using square-based losses will lead a wrong DAG result in structure learning in d-dimensional cases. They also show that scale influences the performance of structure learning if relations among variables are non-linear for both square based and log-likelihood based losses.

**Strengths:**

the authors provide conditions under which using square-based losses will lead a wrong DAG result in structure learning. The conclusions drawn in the paper are reasonable.

**Weaknesses:**

I think that the findings in the paper aren't unexpected. Learning causal structures should go beyond just predictive effects (MMSE). I don't see the paper as particularly significant.

**Questions:**

1.	In this article, MMSE seems to be used for prediction, rather than for learning causal graphs. The definition of MMSE is the sum of squares of predicted residuals, which characterizes the statistical correlation of variables rather than causal relationships. It is an obvious conclusion that only using statistical correlation for causal graph learning will lead to errors.
2.	In structure learning algorithms, the score function usually contains penalty terms, and the variable that needs to be optimized in the score function is the adjacency matrix W, rather than data X. In addition, the adjacency matrix W will also be included in the penalty term. So, why does the score function directly degenerate into a square based loss function? Or, why is optimizing the score function equivalent to optimizing MMSE?
3.	In practical applications, most of the score functions we choose are not affected by data normalization, just like the SRL function mentioned in this article. Thus, this article does not have a clear critical objective, or rather, the critical target seems to be rare in practical applications, which greatly reduces the significance of this study.

---

> ### Author Response · Authors · 2023-11-13
> **Comment by the Authors**
>
> Thank you for your insightful feedback, this will help us to improve our work further. We answer your concerns below.
>
> > In this article, MMSE seems to be used for prediction, rather than for learning causal graphs. The definition of MMSE is the sum of squares of predicted residuals, which characterizes the statistical correlation of variables rather than causal relationships. It is an obvious conclusion that only using statistical correlation for causal graph learning will lead to errors.
>
> We agree that the MMSE characterizes only the quality of some model in terms of how well it fits statistical correlations. However, please note our work considers *structure learning* algorithms which are **not** designed to identify **causal** structures. Rather, these algorithms are supposed to identify **independence** structures within the data at hand. Please note that we explicitly state this in the Introduction section.
>
> > In structure learning algorithms, the score function usually contains penalty terms, and the variable that needs to be optimized in the score function is the adjacency matrix W, rather than data X. In addition, the adjacency matrix W will also be included in the penalty term. So, why does the score function directly degenerate into a square based loss function? Or, why is optimizing the score function equivalent to optimizing MMSE?
>
> Thank you for pointing this out! We agree that scores used for structure learning usually contain penalty terms, e.g. to ensure that the learned graph is a DAG. In our theoretical analysis we assumed for simplicity that the constraints represented by these penalty terms (e.g. graph is a DAG) are fulfilled. Based on that we show that within the space of valid solutions (according to the constraints), structure learners using log-likelihood still may identify wrong structures under certain conditions. We stated this limitation in our work.
> Please note that in our empirical evaluation these penalty terms were **not** excluded, underlining that our theoretical results are still of high relevance in practice.
> Please also note that Proposition 7 shows that any log-likelihood based score is proportional to the MMSE score under Gaussian noise assumption.
>
> > In practical applications, most of the score functions we choose are not affected by data normalization, just like the SRL function mentioned in this article. Thus, this article does not have a clear critical objective, or rather, the critical target seems to be rare in practical applications, which greatly reduces the significance of this study.
>
> This is an interesting point, however we do not agree that most score functions chosen in practice are not affected by data normalization. In fact, our work clearly shows that the entire log-likelihood family under Gaussian noise assumption is affect by scale (see Proposition 8). To the best of our knowledge, most score based structure learning algorithms use such a score as an objective (e.g. GES [1], NOTEARS, [2] DAG-GNN [3], GranDAG [4]). Therefore we believe that our contribution is still significant, also for practitioners.
>
> We hope that we have clarified your concerns and that you can reconsider our rating. Do let us know if you have any further questions or comments.
>
> [1] Chickering. Optimal Structure Identification With Greedy Search. JMLR. 2002.
>
> [2] Zheng et. al. DAGs with NO TEARS: Continuous Optimization for Structure Learning. NeurIPS. 2018.
>
> [3] Yu et. al.  DAG-GNN: DAG Structure Learning with Graph Neural Networks. ICML. 2019.
>
> [4] Lachapelle et. al. Gradient-Based Neural DAG Learning. ICLR. 2019.

---

> > ### Comment · Reviewer_kwyp · 2023-11-18
> > **Thank you for the response**
> >
> > I would like to thank the authors for their thorough response and the clarifications to my questions. I am increasing my score to a 5.
> >
> > In fact, I have another question that I am very confused about. In my understanding, when we using score-based method in the process of structural learning, the scores of all DAGs in the same Markov Equivalent Class should be equal. Why does it occur that the score of one DAG is strictly lower than that of all other DAGs? It seems to contradict my intuition. If I misunderstand, please correct me.

---

> > > ### Author Response · Authors · 2023-11-18
> > > **Thank you for your comment**
> > >
> > > Thank you for considering our rebuttal and for adapting your score!
> > >
> > > Also, thank you for your additional question, we'd like to answer your question as follows:
> > >
> > > First, your intuition is correct that every DAG in the same Markov Equivalence Class (MEC) **should** receive the same score in score based structure learning. However, as we have shown in Proposition 2, assuming Gaussian noise and considering log-likelihood based losses, the scale of the variables dictates which DAG minimizes the loss. For example, consider the graph $X \rightarrow Y$ and assume we obtain data from its induced distribution. As show in Prop. 2 $Y \rightarrow X$ receives a lower loss than the ground truth if Var($X$) > Var($Y$) because the marginal variance of $Y$ instead of the marginal variance of $X$ gets included in the loss as $X$ has an incoming edge (see Prop. 1). Since Var($X$) > Var($Y$) holds by assumption, excluding the marginal variance of $X$ leads to a lower loss than excluding $Y$ from the loss computation, hence $Y \rightarrow X$ is preferred.
> > >
> > > Prop. 5 generalizes this results to MECs and intuitively states the following: If you compare two DAGs $G$ and $G'$ from the same MEC that differ in only one edge orientation, then the DAG containing the edge which is directed from the lower-variance variable to the higher-variance variable is receives a lower loss.

---

> > > > ### Author Response · Authors · 2023-11-22
> > > > **Any further concerns?**
> > > >
> > > > Dear Reviewer,
> > > >
> > > > Thank you again for engaging with us and the score increase to 5. We would like to ask if our follow-up answer resolved your outstanding conern? If yes, it will be great if you can reconsider your rating. Since the discuission phase is coming to a close soon, we would be very happy to answer any of your further concerns in a timely manner.
> > > >
> > > > Regards,
> > > >
> > > > The Authors

---

> ### Author Response · Authors · 2023-11-17
> **Looking Forward to Feedback on Our Response**
>
> Dear Reviewer,
>
> We appreciate the time and effort that you have taken to provide us with your insightful review. We would like to ask if the reviewer has any further concerns or is satisfied by our responses to the original review.
>
> We are looking forward to any further discussion with the reviewer and would like to thank the reviewer again for helping make our paper better.
>
> Regards,
>
> The Authors

---

### Official Review · Reviewer_HNsv · 2023-11-06

**Soundness:** 3 good
**Presentation:** 3 good
**Contribution:** 3 good
**Rating:** 6
**Confidence:** 4

**Summary:**

This work examines the behavior of score based and continuous optimization approaches to causal discovery when mean squared error is used as the objective measure. Building off of the work of Loh & Buhlmann who studied the behavior, and inapplicability, of the MSE under unknown variances, the work examines behavior under non-linear dependence and characterize the failure conditions. The authors then show that under the Gaussian noise assumption all log-likelihood based scoring functions are susceptible to scaling failure conditions. Empirical results are presented on synthetic data which coincide with the theoretical results.

**Strengths:**

This work provides an interesting and important extension in analyzing the issues with applying mean squared error as an objective in structure learning. Given the recent rise in continuous optimization approaches for structure learning I believe this is particularly important. The inclusion of both optimization and scored based methods is also nice. The authors do a good job of clearly defining the problem, and presenting the analysis in a clean manner, along with the implications of results.

**Weaknesses:**

I think the biggest issue here is a lack of discussion of / contextualization with constraint based approaches. This seems particularly important since constraint based discovery still constitutes a large portion of commonly applied discovery algorithms. It would also have been good to see comparisons to constraint based discovery in the experiments.

**Questions:**

Not necessarily a question, but it would be useful if the authors can add a discussion and / or empirical comparison of constraint based discovery methods.

---

> ### Author Response · Authors · 2023-11-13
> **Comment by the Authors**
>
> Thank you for your insightful feedback and we highly appreciate that you agree with us that research on limitations of log-likelihood based scores in score based structure learning is of high importance. We answer your concern below.
>
> > Not necessarily a question, but it would be useful if the authors can add a discussion and / or empirical comparison of constraint based discovery methods.
>
> We agree that a brief discussion on how constraint based structure learners are affected by our results is useful to contextualize our work properly. Therefore we added a discussion on that in our paper, stating that - **as long as the used (conditional) independence test is invariant to scaling the data** - constraint based methods are not affected by our findings. Instead of optimizing a score, constraint based algorithms test for (conditional) independencies among variables and use this information to construct a graph. We performed experiments that confirm our conjecture using PC-algorithm [1] and Grow-Shrink (GS) algorithm [2]. For each algorithm we used the Fisher-Z test.
>
> Our empirical results show that neither PC, nor GS is affected by setting variables to a different scale. The following table shows how often (within 10 repeats) GS predicted a different graph if the data scale changes. The same results hold for PC.
>
> |          | chain | fork | collider |
> |--------|-----|----|--------|
> |   chain  |   0%  |  0%  |    0%    |
> |   fork   |   0%  |  0%  |    0%    |
> | collider |   0%  |  0%  |    0%    |
>
> We will add the empirical results on constraint based methods in the Appendix.
>
> We hope that we have clarified your concerns and that you can reconsider our rating. Do let us know if you have any further questions or comments.
>
> [1] Kalisch et. al. Estimating High-Dimensional Directed Acyclic Graphs with the PC-Algorithm. JMLR. 2007.
>
> [2] Margaritis . _Learning Bayesian Network Model Structure from Data_. Ph.D. thesis, School of Computer Science, Carnegie-Mellon University, Pittsburgh, PA. 2003.

---

> ### Author Response · Authors · 2023-11-17
> **Looking Forward to Feedback on Our Response**
>
> Dear Reviewer,
>
> We appreciate the time and effort that you have taken to provide us with your insightful review. We would like to ask if the reviewer has any further concerns or is satisfied by our responses to the original review.
>
> We are looking forward to any further discussion with the reviewer and would like to thank the reviewer again for helping make our paper better.
>
> Regards,
>
> The Authors

---

> ### Author Response · Authors · 2023-11-22
> **Did we resolve your concern?**
>
> Dear Reviewer,
>
> We hope our provided answer resolved your outstanding concen regarding the constarint based methods.  If yes, it will be great if you can reconsider your rating. Since the discussion phase is coming to a close soon, we would be very happy to answer any of your further concerns in a timely manner.
>
> Regards,
>
> The Authors

---

### Author Response · Authors · 2023-11-13
**Rebuttal**

We thank all reviewers for their thoughtful feedback. This really helped us improving our paper in several regards. We hope that we can address remaining concerns in the answers to each review.

We uploaded a revised version of our paper and marked the changes using the following color coding:

red = Reviewer HNsv

purple = Reviewer FNo4

---

### Meta-Review · Area_Chair_cUTH · 2023-12-14

**Metareview:**

The paper shows that the optimization of a data-fitting loss can be misleading to find the causal structure underlying the data. Basically, a root cause intervenes in the loss through its variance; the optimization objective can thus prefer considering a root cause with high variance as an effect, than as a root cause.

The paper, closely related to the varsortability (Reisach et al, cited), extends the analysis to the non-linear setting.

The paper is good and of good quality. However, as noted by most reviewers, the result is not surprising, which diminishes the merits of the work.

As a suggestion: the paper could be strengthened by comparing the various solutions obtained through varying the norm of the observed variables, and examining the stability of the found SCM through such operations.

**Justification For Why Not Higher Score:**

The paper is warning against a danger that was already well identified.

**Justification For Why Not Lower Score:**

Good quality; appreciated by reviewers; authors took the rebuttal seriously.

---

### Decision · Program_Chairs · 2024-01-16

Accept (poster)